# Targeted V1 comodulation supports task-adaptive sensory decisions

Caroline Haimerl [1,2] ✉, Douglas A. Ruff[3], Marlene R. Cohen[3], Cristina Savin [1,4,6] & Eero P. Simoncelli[1,4,5,6]

Sensory-guided behavior requires reliable encoding of stimulus information in neural populations, and flexible, task-specific readout. The former has been studied extensively, but the latter remains poorly understood. We introduce a theory for adaptive sensory processing based on functionally-targeted stochastic modulation. We show that responses of neurons in area V1 of monkeys performing a visual discrimination task exhibit low-dimensional, rapidly fluctuating gain modulation, which is stronger in task-informative neurons and can be used to decode from neural activity after few training trials, consistent with observed behavior. In a simulated hierarchical neural network model, such labels are learned quickly and can be used to adapt downstream readout, even after several intervening processing stages. Consistently, we find the modulatory signal estimated in V1 is also present in the activity of simultaneously recorded MT units, and is again strongest in task-informative neurons. These results support the idea that co-modulation facilitates task-adaptive hierarchical information routing.

Humans and animals are able to flexibly adapt their behavior according to ever-changing sensory input and goals. In the brain, sensory information is transformed through hierarchical stages of computation, building increasingly complex feature maps[1,2]. However, decisions can rely on local stimulus attributes, which requires not just preserving this information as it ascends the processing hierarchy, but also selecting those aspects of the representation to read out[3]. Consider a decision about local visual orientation. This information is explicitly represented in primary visual cortex (V1), where neurons respond selectively to specific orientations at specific locations in the visual field[4]. However, decisions are not made in V1—visual orientation signals undergo a sequence of transformations, presumably mixing with task irrelevant information (other features of the stimulus or information from other spatial locations) before reaching decision areas. How do areas downstream of V1 access the task-relevant sensory information to flexibly guide behavior?

The problem of flexible sensory decision making has been studied from different perspectives. First, within the traditional "ideal observer" framework, statistically optimal decoders can be constructed from a complete description of response properties of the encoding population, as they pertain to the task. These provide performance upper bounds for behavior[5–11], but fail to explain how a downstream circuit—with limited knowledge of each upstream neuron's stimulus–response and noise properties—can construct such a readout[12]. Second, attentional boosts in the activity of the relevant neurons are believed to highlight task-informative sensory information for downstream processing[13–15]. However, this early-stage encoding selection may be insufficient to ensure the preferential transmission of task-specific information across a complex processing hierarchy[16]. Some have argued that the behavioral benefits of attention are largely due to effective contextual readouts[17], which may explain instances where behavioral-level benefits can be

---

[1]Center for Neural Science, New York University, New York, NY 10003, USA. [2]Champalimaud Centre for the Unknown, Lisbon, Portugal. [3]Department of Neurobiology, University of Chicago, Chicago, IL 60637, US. [4]Center for Data Science, New York University, New York, NY 10011, USA. [5]Flatiron Institute, Simons Foundation, New York, NY 10010, USA. [6]These authors contributed equally: Cristina Savin, Eero P. Simoncelli.
✉e-mail: caroline.haimerl@research.fchampalimaud.org

experimentally dissociated from increases in firing rates[18]. Finally, recurrent dynamics in prefrontal cortex can support context-dependent selection and integration of visual stimuli[19]. This has been demonstrated for cued switching between anatomically segregated stimulus features (such as color and motion), but it is not clear how this mechanism could generalize to the task of making decisions based on different local orientations and in the absence of an explicit cue. We also don't know how the brain could learn the dynamics required for such late selection, from limited task experience.

Here we put forward a theory in which a stochastic modulatory signal induces shared variability in neural responses, which then serves as a label for task relevance. We examine its implications in the context of a change detection experiment in non-human primates[20,21], with blocked task switching. We show that V1 neural responses exhibit fluctuations that can be captured with a shared modulator that preferentially targets task-informative neurons. This task-dependent co-variability acts as a functional label that can be used to guide decoding, and can be learned within a handful of trials, facilitating fast readout from the population. By studying stochastic modulation in an artificial neural network model of the visual hierarchy, we find that the modulatory label can propagate through additional stages of processing, and facilitate readout of task information, with minimal amounts of task-specific feedback. As predicted by the model, the V1 modulatory signal is also present in MT units, again most strongly in those that are task-informative. These results support the hypothesis that task-specific labeling propagates through the visual hierarchy in parallel with stimulus information, facilitating downstream decisions and actions.

## Results

Monkeys were trained to detect a small change in orientation of a Gaussian-windowed drifting sine grating (Fig. 1A), and spiking responses of neurons in their primary visual cortex (V1) and middle temporal area MT were recorded simultaneously (Fig. 1B). Two to three gratings were present simultaneously, at high or low contrast levels, and spontaneously changed their orientation after a variable number of repeated presentations (stimulus on for 200 ms, off for 200–400 ms). The animals were rewarded only for responding to changes of one of these, with the others acting as distractors. The location of the relevant stimulus was fixed within each block of trials, switching randomly between blocks throughout an experimental session. The two possible orientations of the stimulus also switched between blocks. Monkeys were able to quickly adjust to these switches[21], reaching asymptotic performance levels after a handful of trials (Fig. 1C, D). We aim to explain how the brain achieves this impressive combination of accuracy and flexibility.

### Encoding of local visual orientation in a V1 population

Neurons in V1 respond selectively to the local orientation of visual stimuli, and the selectivities of the full population span all orientations and visual field locations, in a topographical organization on the cortical sheet. In the experiment, individual grating stimuli are roughly matched to V1 receptive field (RF) sizes at the eccentricity at which recordings are performed, and orientation changes are relatively small (10–45°, see ref. 21), which restricts relevant stimulus information to a small subset of V1 neurons whose responses change with the stimulus orientation. As nearly all visual information passes through V1[22], the behavior of the monkey must rely on the responses of this subset (the

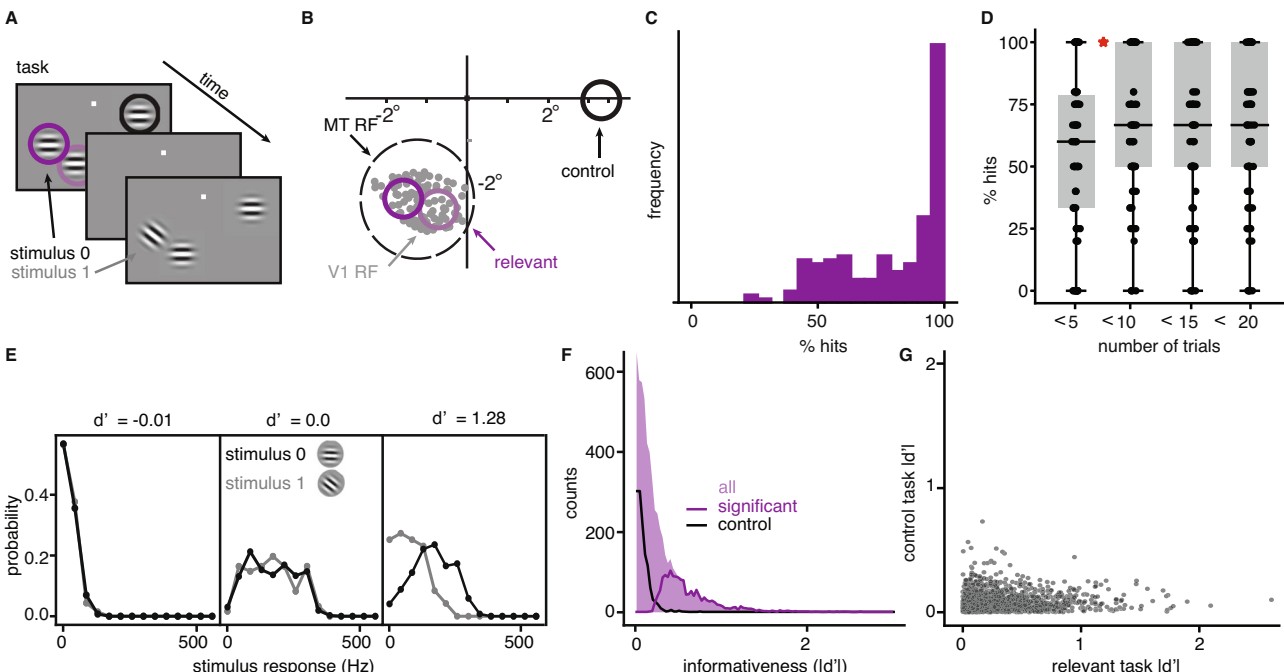

**Fig. 1 | An orientation discrimination task with distractors. A** In each block of trials, 2–3 drifting gratings flash on and off on screen and can change their orientation. One stimulus location is selected as relevant, and the monkey must report changes in its orientation with a saccadic eye movement. **B** The recorded population of V1 neurons has receptive field centers (gray) within the receptive field of a simultaneously recorded MT unit. Two of the three stimuli locations are within the MT unit's receptive field ("relevant"−purple circle, matched to average V1 RF size) and one is in the opposite hemifield ("control"−black). **C** Distribution of behavioral performance across blocks, quantified by the % hits (67 blocks in total). **D** Behavioral performance as a function of time within a block, binned using 5 consecutive trials; the boxes mark 25 and 75% quantiles, horizontal line indicates the median, points indicate different blocks and the red star indicates a significant difference in means (relative two-sided *t* test with 90 test blocks from 24 sessions and 2 monkeys, *p* = 0.015 for pairwise comparison between <5 and <10 groups, *p* > 0.05 for all other comparisons). **E** The distribution of firing rates over all stimulus presentations, to each of the two task stimuli for three example neurons with different *d'* values. **F** |*d'*| distribution, over all blocks of relevant tasks and all V1 neurons (shade). Lines indicate sub-distributions of neurons with significant informativeness (purple), and neurons in the control task (black). **G** Relationship between the informativeness values in relevant and control tasks. **A** and **B** adapted from ref. 21. Source data provided as a Source data file.

same throughout a block), while ignoring the chatter of background activity from the remainder of the population. Moreover, since downstream decision-making areas do not have access to V1 responses directly, the relevant information must be traced as it progresses through various stages of visual processing.

Two of the three stimulus locations were chosen so as to overlap the RFs of the recorded V1 population (Fig. 1A). When one of those locations is task-relevant, we expect a subset of the recorded neurons to provide information for the animal's decision ("relevant tasks"). In contrast, the neurons should be uninformative when the third stimulus location is task-relevant, since it lies in the opposite hemisphere ("control task"; Fig. 1A). We quantified the task-informativeness of each V1 unit as the absolute difference in mean responses for the two orientations relative to response standard deviation ($|d'|$). Figure 1E shows the relationship between informativeness and responsiveness for three representative examples. First, a large number of units are weakly responsive to both stimulus orientations (for instance, because their RFs did not overlap the stimulus location or because their preferred orientation was too different from the relevant stimuli) and consequently cannot be informative about stimulus identity (Fig. 1E, left). Second, some units respond strongly but similarly to both stimuli (Fig. 1E, middle), showing that responsiveness is necessary but not sufficient for task-relevance. Third, some units respond strongly to only one of the two stimuli and hence have high informativeness (Fig. 1E, right). Overall, for each relevant task block, a modest proportion of the recorded V1 units are significantly informative (monkey 1: 25.8%, monkey 2: 18.4%; non-parametric test, see Methods; note that the experimental stimuli are optimized to drive the recorded subpopulation). Only 2.4% and 6% of units are significantly informative in the control task (Fig. 1F). Neurons that are most informative in either of the relevant tasks have low $|d'|$ in the control task, reflecting their task-specificity (Fig. 1G). Across the two relevant tasks, unit informativeness is more similar (61% of significant neurons are informative in both relevant tasks) because of the close proximity of the two relevant stimulus locations. Data sample sizes for each analysis are provided in Supplementary Table S1. The impact of including multiunits in the analysis is discussed in Supplementary Note S9.

Within each task block, a different subset of V1 neurons carries task-relevant information. This subset will be partially overlapping for the two relevant tasks but almost entirely distinct between relevant and control tasks (see Supplementary Note S14). In order to make accurate decisions, a downstream circuit has to read out selectively from those, ignoring the rest. Moreover, the determination of this relevant subpopulation happens quickly: the monkey's performance reaches asymptotic levels roughly 5 trials after each task change (Fig. 1D). How can this flexible routing of information be achieved? Since basic response statistics such as mean or variance do not differ much between informative and uninformative neurons (Supplementary Note S1 and Supplementary Fig. S1A), they cannot guide this selection. Instead, we propose that task-specific structure in the joint statistics of neuronal responses[20,23,24] are key to understanding flexible readout.

## A targeted shared stochastic modulator in V1

Neural responses fluctuate from trial to trial. Some of this variability is neuron-specific, but some is correlated across neurons, driven by circuit dynamics[25–27]. To determine the structure of co-variability, we fitted a modulated stimulus response model ("modulated-SR model") to the recorded population of V1 neurons in each block, using a Poisson latent dynamical system (PLDS, see "Methods" and ref. 28), which jointly estimates the stimulus drive to each unit and the shared, within-trial variability across the population (Fig. 2A, B). The stimulus response component ("SR model") accounts for stimulus-induced transients across multiple time bins of 50ms, with time-specific parameters for each contrast condition (see Methods for details) and

independent Poisson noise. The shared, within-trial variability is assumed to arise from a low-dimensional dynamic stochastic signal, which multiplicatively modulates the stimulus responses of all units, with neuron-specific modulator coupling strengths. This statistical framework allows us to probe the existence, dimensionality, and structure of shared modulation in each block, in a way that simpler dimensionality reduction methods cannot achieve (Supplementary Note S2).

We found that 91% of blocks are better fit by the modulated-SR model than by the SR model alone (Fig. 2C). Moreover, varying the dimensionality of the modulator reveals that 72% of blocks are best described by a one-dimensional modulator (Fig. 2D; see "Methods"). For consistency, we restricted subsequent analyses to these blocks. The extracted modulator is unrelated to contrast variations in the stimulus (Supplementary Note S3) and fluctuates within and across trials at a fairly rapid timescale (Fig. 2B), with no evidence of oscillatory structure. The average estimated time scale of the fluctuations is 75ms (Fig. 2E)—faster than the average trial duration (3s) as well as the individual stimulus duration (200 ms), and approaching the time resolution of spike count binning (50 ms). This fast time scale, together with the unimodal marginal statistics of the estimated modulator (Supplementary Note S4), differentiate it from previously reported on-off dynamics[29].

The improvement in fit quality obtained by including the modulator varies across units (Fig. 2C), but is most prominent in task-informative neurons (Fig. 2F), suggesting that they may be more strongly modulated. A non-parametric comparison revealed that task-informative neurons have larger coupling weights (i.e. stronger modulation) than uninformative neurons (Fig. 2G). Although informativeness is correlated with the mean firing rate of a unit (Supplementary Note S5), a partial correlation analysis confirmed that firing rate differences cannot explain the inferred modulation targeting, as firing-rate-corrected informativeness and modulator couplings are significantly correlated in 84% of blocks (Spearman $r$, $\alpha = 0.05$; Fig. 2H–J). The increased variability in the task-relevant neurons (Supplementary Note S1) is primarily due to the modulation; residual variability unexplained by the modulated-SR model is generally not correlated with informativeness (Spearman $r$ with $\alpha = 0.05$; Fig. 2J); only 9% of blocks have significantly positive correlations between residual variability and informativeness (19% significantly negative). While most of this residual variability is neuron-specific, we also find weak, structured correlations in pairs of units which suggest additional sources of shared noise not captured by the model (Supplementary Fig. S2).

The modulator coupling is dissociable from traditional attentional effects on mean firing rate (Supplementary Note S7), which have been suggested to improve encoding precision of particular attended stimuli[30], and it cannot be explained by neural adaptation, as the degree of adaptation was uncorrelated with the quality of the fit of the modulated-SR model (Supplementary Note S8). Finally, the modulator structure cannot be explained by the fact that the response measurements are in the form of multiunit spike counts (Supplementary Note S9). Overall, our analysis reveals that V1 responses are modulated by a common fluctuating signal, and that the strength of this modulation in each unit reflects its task-informativeness. From an encoding perspective, this seems counter-intuitive (Supplementary Note S10). Why would the brain inject noise specifically in the few neurons that matter most?

## Targeted modulation can facilitate decoding

The modulator fluctuates rapidly, allowing any task information it provides to be accessed quickly, potentially on the time scale of single trials. We hypothesize that the modulation serves to "label" the responses of the task-relevant V1 subpopulation, so that downstream circuits can easily identify and use these signals.

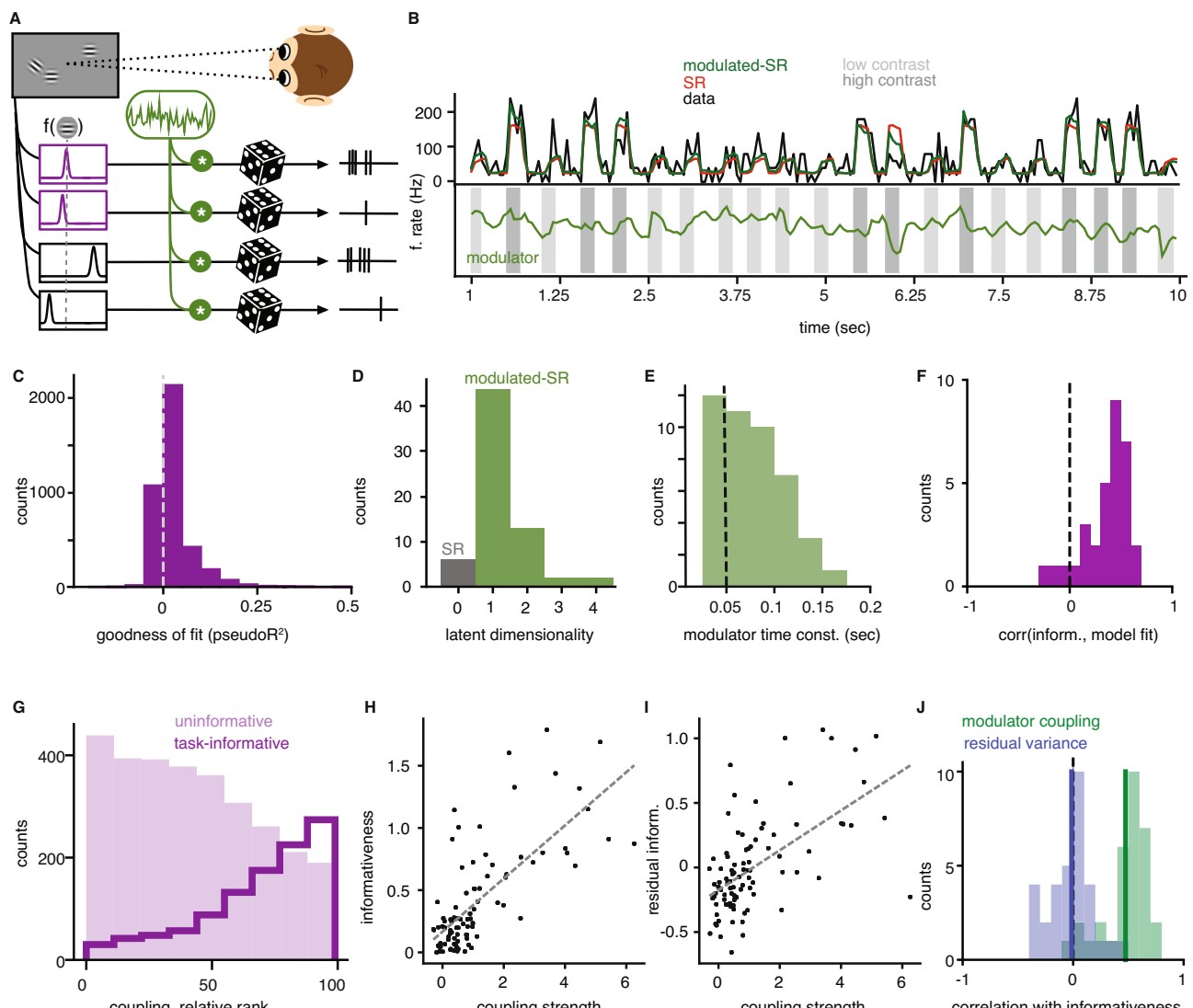

**Fig. 2 | Estimating V1 modulator. A** An illustration of the modulated stimulus response model: Each neuron's tuning function is modulated by a time-varying shared source of multiplicative noise (green), with spiking modeled by a Poisson process (image adapted from Haimerl et al.[12]). **B** An example unit's activity over concatenated test trials of a block and the corresponding prediction of the SR model and the modulated-SR model. Bottom row shows the estimated trajectory of the modulator. **C** The distribution of pseudo-$R^2$ values over all neurons in blocks that were best fitted by a 1-dimensional modulated-SR model. **D** Summary of the dimensionality of best fitted models across relevant tasks. **E** The distribution of estimated time constants over all blocks that were best fitted by a one-dimensional modulated-SR model. **F** Distribution of the correlations between the individual unit's model fit (pseudo-$R^2$) and their informativeness (78% of blocks have significant positive correlations between informativeness and model fit, measured via a two-sided Spearman $r$, $\alpha < 0.05$). **G** Relative population rank of modulator coupling strength (within each block) for significantly informative (dark purple line) and uninformative (light purple shading) neurons. **H** Informativeness vs. coupling strength for an example block. **I** Residual informativeness (unexplained by linear effects of mean firing) vs. coupling strength in same example as (**H**). **J** Distribution of correlation coefficients obtained by partial correlation analysis across blocks (green, 84% of blocks significant two-sided Spearman $r$) and a similarly obtained distribution that uses the modulated-SR model residual response variance as a proxy for neuron individual variance (blue). Source data are provided as a Source data file.

To analyze the decoding process, we simulated an encoding model that captures the essence of the response properties observed in the V1 data. For this, we use a variant of the modulated-SR model with static stimulus-dependent firing rates, and one shared, temporally-independent stochastic modulator $m_t$ (see "Methods," and ref. [12]):

$$k_{n,t}(s) \sim \text{Poisson}\left(\lambda_n(s) \exp(c_n m_t)\right), \tag{1}$$

where $k_{n,t}(s)$ is the spike count of neuron $n$ at time $t$ in response to stimulus $s$; the modulator $m_t$ is drawn independently from a Gaussian distribution with zero mean, and influences neuron $n$ with coupling weight $c_n$, which is set to be proportional to the neuron's task-informativeness. Finally, since the degree of modulation affects not only variability but also mean responses, we explicitly correct for the mean increase to isolate the effects of modulator-induced co-variability (see "Methods").

Given this encoding model and a binary discrimination task, $s \in \{0, 1\}$, the ideal observer's optimal decoder compares a weighted sum of the neural responses with a modulator-specific decision threshold, $q(m_t)$ (see "Methods"):

$$\sum_n a_n^{(\text{opt})} k_{n,t}(s) > q(m_t), \tag{2}$$

where $a_n^{(\text{opt})} = \log(\lambda_n(1)) - \log(\lambda_n(0))$ denotes the optimal decoding weights. These are independent of the modulator and equivalent to

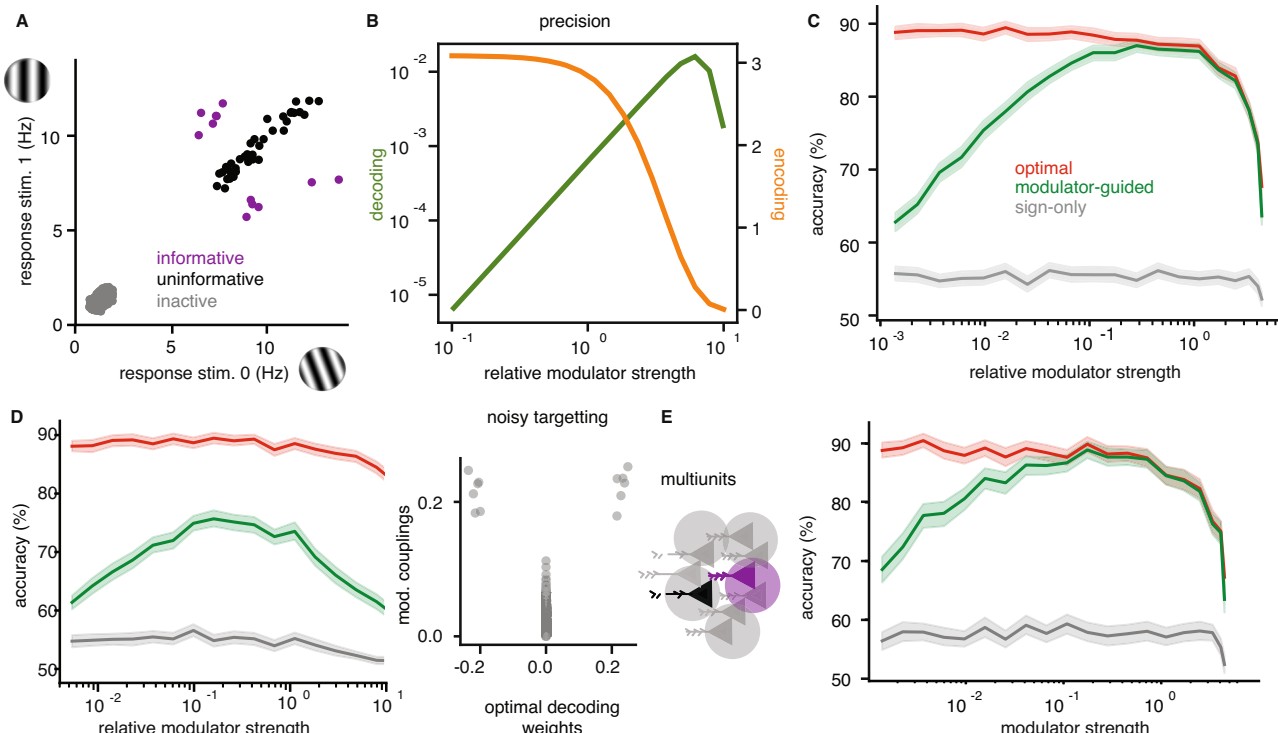

**Fig. 3 | Theory of modulator guided decoding. A** The average response of neurons of the three subpopulations to two task stimuli. There are 12 informative, 38 uninformative and 4950 inactive neurons. **B** Effects of increasing modulator strength on encoding and decoding, respectively, for modulator coupling weights equal to informativeness. Encoding is measured by the SNR, while decoding precision is quantified as the variance of the decoding weights of the modulator-guided decoder. **C** Performance of three different decoders in simulations of a discrimination task with 1000 model V1 neurons, 50 informative, with increasing relative modulator strength (mean and 95% confidence interval). **D** Same comparison as in (**C**), but with the modulator coupling weights corrupted by Gaussian noise, as shown in the right panel. **E** Same comparison as in (**C**), but with simulated multiunits, obtained by summing the activity of random pairs of neurons.

those derived from an independent Poisson model. The decoding weights are non-zero only for the small subpopulation of informative neurons (Fig. 3A, purple), with their signs indicating preference between the two stimulus alternatives. Zero weights eliminate active but uninformative (Fig. 3A, black) or inactive (Fig. 3A, gray) neurons.

The optimal decoder provides an upper bound on decoding performance given the encoding model, and motivates the use of a linear-threshold functional form for the readout. However, it uses weights that rely on full knowledge of each neuron's mean responses to the stimuli of the current task. The challenge for a downstream circuit is to find a way to approximate these weights, when provided only with incoming spikes, the task feedback, and potentially the modulator, but without explicit knowledge of the stimulus encoding model. How can the brain achieve this? The conventional means of learning decoding weights is regression. Although this is feasible for a small set of mostly informative neurons, the number of training examples needed for accurate weight estimation grows significantly with population size[31,32]. So the behavioral flexibility exhibited by the monkeys precludes such a solution. Instead, we seek a heuristic that can be estimated quickly.

Consider first a decoder motivated by early work on neural binary discrimination[33]. The idea is to split all neurons into two subpopulations ("preferred" and "anti-preferred") and then compare their average responses. This solution only assigns decoding signs ($a_n^{SO} \in \{-1,1\}$), which indicate relative stimulus preference, but ignores the relative importance of different neurons (there are no zero weights); we refer to this approach as the *sign-only* (SO) decoder. It can be learned quickly (Supplementary Note S10), but its performance falls as the fraction of informative neurons decreases (Supplementary Note S10): Since all neurons must be included, the noise from the uninformative neurons corrupts the decision signal. For realistically

small fractions of informative neurons[5,30], the SO decoder cannot match monkey performance (Supplementary Note S10).

To improve performance, the readout needs to consider the relative importance of individual neurons. A decoder can achieve this by estimating the amplitude of individual decoding weights. Since the relative strength of modulation of each neuron reflects the relative informativeness (by design $c_n \propto |d'|$), we can define a *modulator-guided* (MG) decoder that sets its decoding weight amplitudes from temporal correlations of the modulator with each neuron's activity, which provide a simple estimate for $c_n$:

$$|a_n^{(MG)}| \propto \frac{1}{T}\sum_t m_t k_{n,t}(s). \tag{3}$$

The MG decoder does not rely on knowledge of the response properties of the encoding population, but it assumes access to the modulator (e.g., it is a broadcast signal). This has important implications for learning the decoder; the MG weight estimates converge rapidly, on the time scale of the modulator fluctuations which are much faster than a trial (see "A targeted shared stochastic modulator in V1"). Once the informative neurons have been identified, their decoding sign is determined based on explicit trial feedback, which only requires a handful of trials for small populations (Supplementary Note S10). For simplicity, the amplitude and sign were estimated separately here. Nonetheless, the two can also be learned jointly using a form of local online learning based on eligibility traces[34,35] (Supplementary Note S11).

We compared the performance of different decoders in a binary discrimination task, based on simulated responses of a large population of V1 neurons with a small fraction of informative neurons (5%, Fig. 3A; see also Supplementary Note S10D for variations in percentage

of informative neurons). The statistically optimal decoder corresponds to the ideal observer's solution, and thus provides an upper bound on achievable performance; the SO decoder provides a lower bound. The optimal decoder's accuracy deteriorates as the modulator increases in amplitude, corrupting the encoded signal (Fig. 3C). This reinforces the point that, unlike other forms of noise correlations[32,36], the targeted, multiplicative noise is strictly detrimental for encoding (Supplementary Note S10). While the performance of the MG decoder is limited by this corruption as well, it also benefits from a stronger label in the informative neurons (Fig. 3B). Its performance follows an inverted U-shape as a function of modulation amplitude, reflecting the trade-off between these two opposing effects (Fig. 3C). MG decoding performance is maximized at an intermediate modulation amplitude, where it attains an accuracy close to that of the ideal observer, a result that is robust to variations in population size (Supplementary Note S10). Finally, the MG decoder outperforms classical regression-based approaches in their speed-accuracy trade-off, allowing for quick learning with limited data and asymptotically near-optimal performance in large datasets (see Supplementary Note S10 and "Learning modulator targeting in a hierarchical circuit").

In practice, the performance of the MG decoder could depend on how strongly correlated the modulator couplings, $c_n$, are with task-informativeness. To test the robustness of the MG decoder, we weakened the correlation between the modulator couplings, $c_n$, and task-informativeness by adding noise to $c_n$. We found that although performance decreases overall, the nonmonotonic dependence of the MG decoder performance on modulator strength is preserved (Fig. 3D). Given that our measurements mostly include multiunits, we also tested their impact on decoding and found that the results are qualitatively robust to such measurement noise (Fig. 3E). Interestingly, the optimal modulation amplitude generally shifts towards the range estimated from the data, suggesting that physiologically, the degree of modulation may be well-matched to the precision of the modulator targeting.

## V1 modulator is task specific and facilitates decoding

In our experimental context, the theory predicts that the co-variability of neural responses should change based on whether they are task-informative. Given that the recorded V1 population is informative in the relevant tasks but not the control task (Fig. 1G), we expect differences in overall modulator strength across tasks and in individual modulation strengths across neurons. Indeed, the overall strength of the estimated modulation significantly decreases in the control task, both in absolute terms and relative to stimulus induced variations (Fig. 4A and Supplementary Note S12). In comparison, the two relevant task conditions have indistinguishable statistics of overall modulation strength (Fig. 4B). Our theory explains this difference as a change in labeling, from the recorded subpopulation that is informative for the relevant tasks, to a different (unrecorded) subpopulation that is informative in the control task.

The comparison between the two relevant tasks is limited by the proximity of the two relevant stimulus locations, as only few units are exclusively informative in one task (see "Encoding of local visual orientation in a V1 population"). However, despite the reduced sample size, we find a significant correlation between the difference in informativeness in the two relevant tasks and the difference in coupling (Spearman correlation, $r = 0.16$ with $p < 0.05$), showing that units that are more informative in one of the two tend to also have higher coupling in that task.

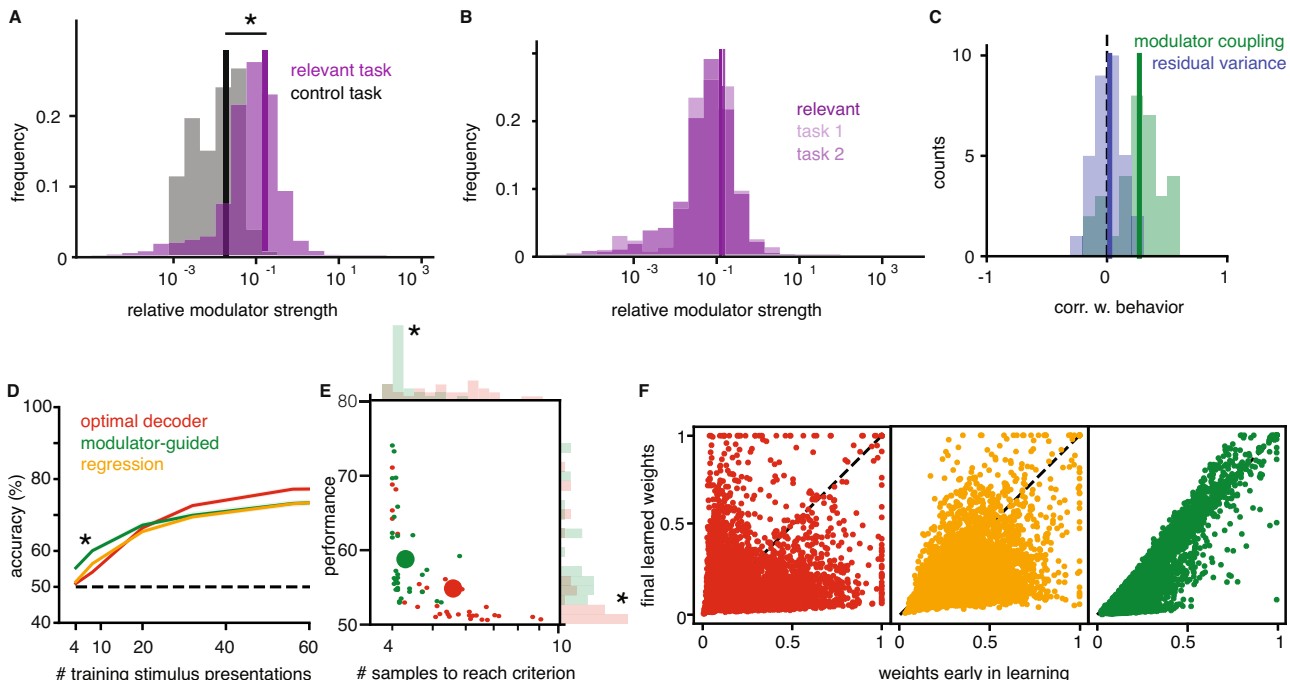

**Fig. 4 | V1 modulator is task-specific and facilitates decoding. A** The distribution of relative modulator strengths across all relevant task blocks (purple) and all control task blocks (black). The star indicates significant difference between the two distributions (two-sided $U$-test, $p = 0.0008$). **B** Same as (**A**), but comparing across the two relevant tasks (two-sided $U$-test, $p = 0.45$). **C** The distribution of correlation coefficients between modulator coupling (green) or residual response variance (blue) and the residual behavioral relevance of a unit's activity (correlation with behavior), obtained by regressing out informativeness and mean firing rate. **D** Decoding from the recorded V1 population: performance of different decoders or logistic regression for an example block population with increasing number of training samples (mean ± SEM); star indicates significant differences between the optimal and the MG decoder (two-sided $t$ test, $p < 1e - 12$). **E** Performance with minimal training against minimal number of training samples (stimulus presentations) needed to reach above chance (50%) performance, for each block; stars indicate significant differences between the optimal and the MG decoder (two-sided $t$ test, $p < 0.0001$ for minimal training, $p = 0.0116$ for performance). **F** Decoding weights estimated with maximum training (90% of all stimulus presentations) versus with minimal training (1%) for various decoders; same colors as (**D**, **E**). Source data are provided as a Source data file.

In our framework, decoding weights are approximated by estimating coupling strengths, and thus neurons with large coupling (and thus strongly modulated) should have a stronger influence on behavior. Despite V1's early position in the visual processing stream, we find this to be true in our data; 91% of blocks show significant correlations (Spearman $r$, $\alpha = 0.05$) between modulator coupling and a unit's correlation with the monkey's behavior computed as a $d'$ of neural responses, with categories defined by the animal's choices rather than stimulus identity (see "Methods"). Potential confounds in this analysis are not only overall firing rates, but also the informativeness of a unit, as the most informative neurons would be expected to have a stronger influence on behavior[37,38]. Nonetheless, even after controlling for these confounds, it remains the case that units that are more modulated are the ones that are also more predictive of behavior (Fig. 4C). This relationship is not present for the residual response variance (Fig. 4C). Furthermore, we do not find a relationship with behavioral correlation in other shared noise sources (Supplementary Note S13), which suggests that the shared modulator-induced fluctuations are particularly relevant for downstream processing.

The most direct prediction of the theory is the ability of the MG decoder to set appropriate decoding weights for the recorded V1 responses, and to do so rapidly, with limited data. To test these predictions, we decoded the stimulus identity from V1 responses using our heuristic MG decoder and compared its performance with that of the ideal observer for the estimated (modulated-SR) encoding model. When all available data is used for estimation, the MG decoder performance is close to that of the optimal decoder (~80% correct, which suggests that the strength and targeting precision of the estimated modulator is sufficient to guide decoding).

The optimal decoder provides an upper bound on decodability assuming perfect knowledge of the V1 response properties, but it can still perform poorly when the model is estimated from limited data; in fact, its performance is at chance in the low-data regime (Fig. 4D). Similarly, learning decoding weights directly through logistic regression requires many training trials before performing above chance (Fig. 4D). In contrast, the modulator-guided (MG) decoder finds informative units after only a few training examples, as it estimates the modulator coupling on the time scale of the modulator itself instead of that of trials. It outperforms the learned optimal decoder and logistic regression in the small training sample regime (comparing MG against either learned optimal or regression-based decoder significant; $t$ test, $p < 0.0001$, see Fig. 4D). We quantify this effect across all data and find that the MG decoder reaches above-chance performance significantly faster than the learned optimal decoder ($t$ test, $p < 0.0001$, Fig. 4E) and that the performance attained with minimal training is significantly higher relative to that of the learned optimal decoder ($t$ test, $p = 0.01$). The MG decoder also reaches above-chance performance significantly faster than a regression-based decoder ($t$ test $p < 0.001$) and learned optimal and regression-based decoder do not differ significantly ($t$ test, $p > 0.05$ for minimal training and performance). A different approach would be using support vector machines (SVM) which are known to provide good weight estimation for limited data. Indeed, an SVM decoder performs similarly to the MG decoder on our data, although it lacks biological plausibility as a decoding mechanism of the brain (see Supplementary Fig. S1E, F). Our theory predicts that the advantage of the MG decoder lies in its ability to accurately estimate the decoding weights quickly. Indeed, we find a strong correlation between the MG decoding weights obtained with minimal training and those estimated from all available data, but this relationship does not hold for the learned optimal decoding weights or the regression weights (Fig. 4F).

Although significant, the difference in the number of trials required for above-chance performance may seem small. Nonetheless, it is likely that the benefits of modulation are substantially underestimated due to two experimental limitations. First, the recorded

subpopulation is biased towards informative neurons since the stimuli are placed so as to drive these neurons. The animal must decode the information present in the entire V1 population, with a much lower percentage of informative neurons. Under such conditions, finding the few informative neurons from task feedback becomes even harder (Supplementary Note S10), and the benefits of modulation stronger. Second, the modulator may vary on a time scale faster than the stimulus-presentations of the experiment and model, which would allow an even faster estimation of the decoding weights (Eq. (3) could also be applied to single spikes). Finally, we found additional sources of co-variability not considered in the theory (measured as residual pairwise correlations, see Supplementary Note S6) which are consistent with previously documented effects of the task condition noise correlations[21]. These do not seem to interfere with the ability of the targeted modulator to facilitate decoding, suggesting that the theory is robust to deviations from the exact model assumptions. Overall, the benefits of the MG decoder for the V1 data provide strong support for the hypothesis that the brain could use task specific modulation to enable flexible task switching.

### Learning modulator targeting in a hierarchical circuit

Visual information processing is hierarchical, and task-relevant information needs to propagate through several stages before reaching decision-making areas. Moreover, since receptive field sizes increase across stages of processing[1], localized task-specific information will diffuse in subsequent visual layers, making the task of identifying the subpopulation of relevant readout neurons even harder. Thus, the decoding problem identified in V1 persists, and likely worsens, in downstream areas. As a separate issue, while thus far we have assumed the correct modulator targeting to be already present in the circuit, the right degree of modulation for each neuron in a task needs to also be learned from experience. Can the modulator-guided readout still facilitate flexible and accurate task performance under these conditions?

To answer this question, we use an artificial neural network to model the visual processing hierarchy with a stochastic modulator and learned targeting. The first layer of the network consists of a V1-like encoding population with localized oriented filters, whose responses are then propagated through two processing layers of neurons with increasing RF size, and finally read out by a decision stage (Fig. 5A; details in "Methods"). To reflect previous experience, connections between stages are pre-trained (via backpropagation), to solve a general image classification task (identifying handwritten digits[39] randomly positioned in different locations; Fig. 5B, C), in the absence of the modulator. As a result of this optimization, the model is capable of discriminating complex visual features.

Analogous to the V1 experiment, we use stochastic modulation to fine-tune this network to the task of discriminating the orientation of local gratings (Fig. 5D, E). After adjusting the decision circuit to the new data (see Methods for details), the network needs to perform a binary discrimination task involving two orientations at a fixed location (Fig. 5E). As in the actual experiment, distractors are placed at other locations in the image, something which the network has not encountered during the previous episodes of learning. We introduce shared, stochastic gain modulation with neuron-specific coupling parameters in the primary encoding layer of the network (with the same functional form as the original encoding model in Eq. (1), but without the Poisson noise; see "Methods" for details). This injected variability accompanies the stimulus information across the processing layers. The responses of neurons in the last layer are combined with gain terms $g_n$, which tune the readout of the decision circuit to the specific task (Fig. 5D). As for the MG decoder in Eq. (3), these gains are adaptively computed using the correlations between the individual neural responses and the modulator, which is again assumed to be available at the decision stage. We optimize the modulator coupling

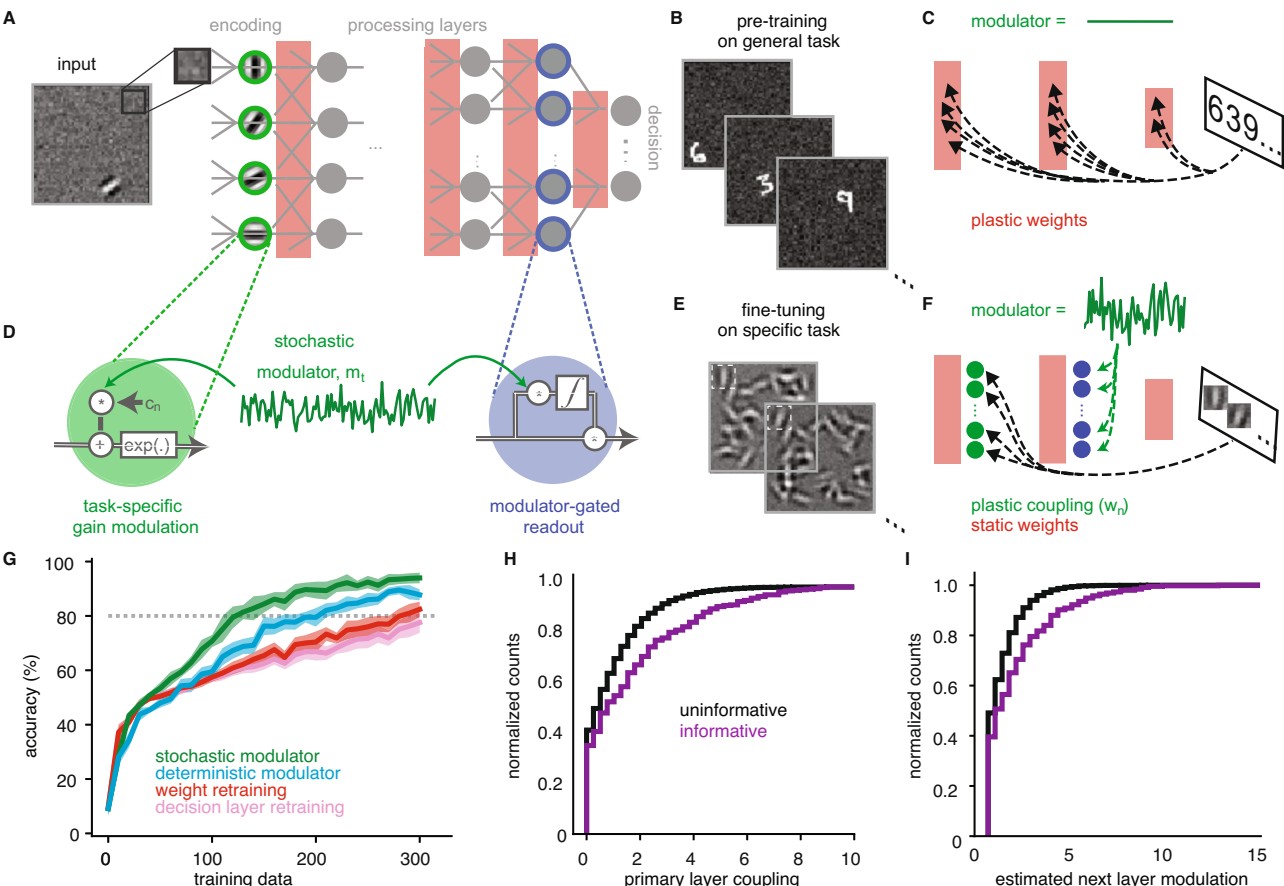

**Fig. 5 | Learned stochastic modulation in a hierarchical network. A** Network with a primary encoding layer consisting of 2560 neurons with fixed Gabor filters with varying orientation and RF location, two locally connected processing layers and a fully connected decision layer. **B** Pre-training on a spatially invariant version of the classic MNIST classification. **C** Pre-training involves optimizing weights of the processing and the decision layer. **D** A stochastic modulator $m_t$ varies the gain of each primary layer neuron according to their coupling terms $c_n$ (green). The fluctuations introduced at the primary layer guide the gain term of the input to the decision layer through a modulator-gated learning rule (blue). **E** Task training involves binary discrimination of grating orientation at a particular location in the presence of distractors. **F** Task-training involves learning the coupling terms $c_n$ via task feedback and adjusting the modulator-gated readout accordingly. **G** Performance of different decoding strategies, as a function of the amount of data used for task training (shown are mean and 95% confidence interval). Gray dotted line indicates criterion performance. **H** Distribution of task-optimized modulator coupling for most informative neurons (5% highest $|d'|$ values) vs. all other neurons at the primary encoding layer. **I** Estimated neuron-specific modulation at the first processing layer for most informative neurons vs. the rest.

strengths to maximize behavioral performance on the task, using explicit trial feedback (via backpropagation). The general rationale is that if task-informative neurons can be modulator-labeled in the V1 stage, then this labeling will be inherited downstream by exactly those neurons that receive their signal. Thus their co-variability can guide decoding at the decision layer.

We assess the efficiency of the modulator-based solution by comparing it to two alternative models, both of which adapt based on experience within the task, but which differ in their parameter complexity. At one extreme, we consider a system that relearns the connection strengths between all layers de novo ("retraining"). This approach corresponds conceptually to the regression model in Fig. 4. At the other extreme, we consider a fixed network that only relearns the final readout weights ("readout only"). Retraining all network weights requires many training examples to reach good performance (defined as > 80% accuracy; Fig. 5G), likely due to the high dimensionality of the parameter space. Retraining only the decision layer results in poor performance, because the presence of distractors renders the pre-trained representation insufficient for effective category discrimination. Compared to alternative models, fine-tuning the network via the modulator substantially reduces the amount of task-training required to reach criterion performance (Fig. 5G).

The improvement in performance of the modulator solution over regression-based relearning corresponds qualitatively to what we found when decoding from the data in Fig. 4D). Nonetheless, one important distinction between this hierarchical model and the previous MG decoder is that the modulator affects both the mean and the variance of the V1-like encoding layer (see "Methods"). To disambiguate the effects of modulation on neural variability vs. mean responses, we introduce a third model, which is parametrized and trained in the same way, but deterministically boosts the gain of initial stage neurons[16], in the absence of stochastic modulation. We find that targeting of deterministic gain modulation can be learned faster than retraining all the connections, but it does not reach the same performance as the stochastic modulator given limited training. This suggests that the separation of stimulus information and task relevance into the mean and variance of neural activity, respectively, further enhances the identifiability of the stimuli at the decision stage.

When investigating the properties of the learned solution, we find that the learned couplings are highest for task-informative neurons (5% highest $|d'|$) in the primary encoding layer (Fig. 5H), as in the data (2F–J). Although the modulator only affects the responses of these neurons directly, we find that informative neurons in the downstream processing layer are also preferentially correlated with the modulator

(Fig. 5I). This suggests that task relevance propagates along the hierarchy in parallel to the stimulus information.

## Modulator label is preserved in MT activity

The model predicts that task-specific modulation introduced in V1 should label task-informative neurons in downstream areas. We look for signatures of such labeling in simultaneously recorded MT activity. MT neurons are known to receive direct input from V1[40] and selectively combine these afferents to construct their receptive field properties, such as motion selectivity[1,41]. Their receptive fields are larger and more complex, responding to localized gratings with different combinations of position, speed and orientation[41,42]. Given anatomical considerations, we expect correlated activity in V1 to drive MT to some extent. What is specific to our theory is the prediction that the degree of inherited modulation should reflect the task informativeness of individual MT units.

We find that responses of MT units that cover the two relevant stimulus locations (Fig. 1A) vary in their task-informativeness (Supplementary Fig. S6A) and show different degrees of supra-Poisson variability (Fig. 6A), suggesting different levels of modulation[43]. The two measures are correlated across the MT units, with informative units having higher Fano factors (correlation coefficient of 0.48, $p < 0.008$). To test whether the excess variability arises due to V1 modulation, we compared two models of MT activity. The first is based on the visual stimuli alone ("SR"); it resembles the V1 SR model, but includes stimulus drift direction (consistent with previous literature[41],

drift direction did not have predictive power for the V1 units, see also[21], but has a strong effect on MT activity). The second model additionally conditions on the modulator estimated from the simultaneously recorded V1 units ("SR+V1 modulation"; Fig. 6B). The SR model provided a good fit for all MT units (Supplementary Fig. S6A), which is expected given that experimental stimuli were optimized to drive MT units. The inclusion of the V1-estimated modulator improved the fit for 73% of the MT units (measured as difference in pseudo-$R^2$, see "Methods"; Fig. 6C). This effect is preferentially observed in task-relevant units, which show a significantly larger model fit improvement relative to the uninformative units (t test, $p = 0.01$; Fig. 6D). Interestingly, this relationship was present only if the estimated V1 modulator showed significant targeting structure (significant Spearman correlations between coupling and informativeness); the few outlier blocks without structured V1 targeting could not explain MT variance (Supplementary Note S16).

The fact that both V1 and MT units are co-modulated as a function of their task informativeness is consistent with our theory, but does not exclude alternative patterns of information flow, such as top–down influences of MT on V1, or independent modulation of both areas from an external signal. To more directly address the nature of the modulation in MT we take advantage of a smaller set of MT population recordings (partly published in ref. 21). Despite the technical differences in recording procedure, this data recapitulates the same overall statistics, with 60% of the MT units having a significant part of their variability explained by the V1-estimated modulator.

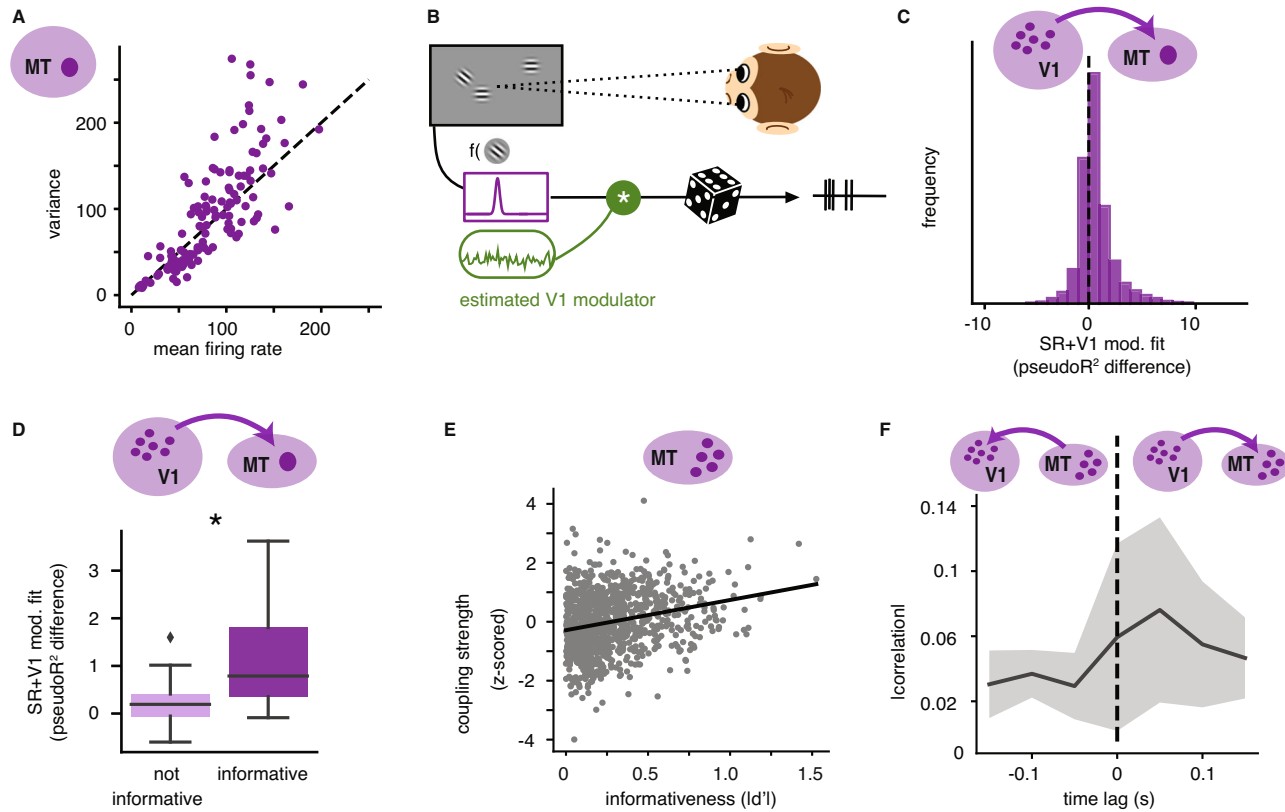

**Fig. 6 | Effects of V1 modulator on simultaneously recorded MT units.**
**A** Stimulus response variance as a function of mean firing for all MT units, and stimulus presentations. **B** Schematic of the model; the spiking of each MT unit is specified by a tuning function potentially multiplicatively gated by the modulator estimated from V1 activity, with Poisson noise (image adapted from Haimerl et al.[12]). **C** Distribution of model fit (pseudo-$R^2$) values obtained by comparing the log-likelihood of the SR model that includes the V1 modulator as an additional dimension (SR+V1 modulation model) against the SR model. **D** Improvement in fit quality for the SR+V1 modulation model, grouping MT units into those with high

informativeness values (50% with highest $|d'|$) and those uninformative (total of 30 MT units). Boxplot shows median and interquartile range. Black star indicates significant difference (two-sided t-test, $p = 0.01$). **E** A modulator is extracted from a population of MT cells. Shown are modulator couplings over informativeness in MT units over all 43 blocks. **F** Correlations of the extracted V1 and MT modulators with positive (V1 before MT) and negative (MT before V1) time lag in seconds. Data from 4 blocks in 4 different sessions of one monkey, see also Supplementary Table S1. Source data are provided as a Source data file.

When independently extracting a modulator from the joint MT population responses ("SR+MT modulation"), we find that this population model better explains individual unit responses than the SR model (in 72 out of 73 blocks, Supplementary Note S17). The extracted modulator has consistent statistics across stimulus contrast variations (in 72% of blocks; Supplementary Note S17) and has similar time constants as those separately extracted in V1 (mean 61 ms, s.d. 20 ms). Lastly, there is a significantly positive correlation between modulator coupling and informativeness across blocks (Pearson $r = 0.24$, $p < 0.0001$, Fig. 6E), suggesting that the same kind organization seen in V1 is qualitatively replicated in MT responses. Are these properties inherited from V1? We find that the cross-correlogram of the V1 and MT-extracted modulators is maximal at a time lag that is consistent with feedfoward propagation from V1 to MT (Fig. 6F), although additional data and finer temporal precision will be required to more definitively understand this relationship. Altogether, our analysis of MT responses supports the idea that the modulation of task-relevant neurons in V1 is passed on to task-informative neurons in MT, allowing the propagation of labeling information towards decision areas.

## Discussion

Humans and animals are impressive in their ability to respond rapidly and precisely to a variety of sensory stimuli, but the neural mechanisms supporting this flexibility remain poorly understood. We have presented a theory for flexible information readout, in which a modulatory signal induces shared response fluctuations in task-relevant cells, accompanies the task-relevant information as it propagates through subsequent stages of neural processing, and facilitates accurate decisions. We uncovered evidence for this labeling scheme in neural recordings from primate areas V1 and MT, obtained while the animals switch between local orientation discrimination tasks at different spatial locations. In particular, targeted modulation in V1 is sufficient to decode stimulus identity from neural responses after observing only a few trials. We also found evidence for the propagation of this modulator to informative neurons in downstream area MT.

The computational challenges faced by downstream circuits involved in decoding have been explored in seminal work by Shadlen and colleagues[33], who enumerated three potential factors that could reduce an animal's behavioral performance compared to predictions of an optimal decoder (the "ideal observer") operating on a hypothetical population of independent neurons: "*suboptimally stimulated neurons*" (in which the decoder includes irrelevant neurons in computing its decision), "*correlated noise*" (which worsens performance since it cannot be averaged out by the decoder), and "*pooling noise*" (additional noise in downstream circuits, whose contribution appears to be small[44]). The first factor has likely been underestimated in experimental data, since the recorded subset of neurons are typically not representative of the full population. As such, our conclusions regarding the benefits of targeted modulation for downstream readout are likely understated. The second factor, correlated noise, can either facilitate or impede stimulus encoding[26]. In particular, differential correlations, such as those reported in mouse V1[45], are information-limiting. They restrict the encoding benefits that would otherwise arise from increasing population size[36] (but might support coding robustness[46]). Irrespective of correlation structure, identifying appropriate decoding weights using regression requires many trials[47], so flexible decoding remains a problem. In contrast, although our modulator-induced correlations are also information-limiting, their robustness to averaging enables the propagation of task relevance labels. Furthermore, their rapid time scale allows for fast estimation of task-specific readouts. Finally, the changes introduced via the modulation are task-specific and ephemeral, allowing the circuit to instantly disengage from the task and revert to its original state, by simply reducing the strength of the modulator.

Top-down attention can facilitate sensory encoding, and has been shown to selectively affect neural responses, including increases in mean response[13-15], decreases in response variability[48], and decreases in noise correlations[30,48,49], all of which increase the signal-to-noise ratio (SNR) of the local sensory representation. These benefits for encoding are distinct from the modulatory effects we have explored here. They operate on the time scale of task conditions (minutes) or stimulus presentations (seconds), whereas the modulator that we estimate here fluctuates on a time scale of tens of milliseconds or faster. In addition, while attentional gain boosts are tuning-specific[49-51], we do not find evidence that they are specific to task-informative units (Supplementary Note S7). Moreover, the estimated modulator coupling is unrelated to the strength of attentional changes of the mean, suggesting that it may arise from separate mechanisms. This is consistent with effects of superior colliculus (SC) inactivation[18], and results documenting a similar dissociation between increases in mean and improvements in behavior over learning in V4[52]. In the context of our theory, we hypothesize that SC inactivation may selectively disrupt the strength or targeting of modulation, affecting the propagation of task-relevant information to decision areas, a prediction that can be tested experimentally.

Our modulator is distinct from slow multiplicative, low-dimensional noise reported in other contexts[53,54], which may serve other functional roles such as encoding uncertainty in visual areas. It is also distinct from gain changes due to fluctuations in attention which operate on the time scale of seconds[55]. Such signals are too slow to serve as a labeling mechanism of the type proposed here. Choice-related feedback signals have also been shown to modulate neural activity on a trial-by-trial basis, but they also occur on a slower time scale of several hundreds of milliseconds or seconds[24,56]. All of this suggests that the modulatory process of our theory does not replace, but coexists with these additional forms of gain modulation.

Shared oscillatory structure induces low-dimensional covariability and has been proposed as a mechanism for binding information across neurons[57]. The "communication through coherence" (CTC) theory[58,59] formalizes this idea in an encoding-decoding framework, in which a top-down oscillatory modulator projects to both encoding neurons with the same feature selectivity, and to the decoding network that needs to read them out. The modulators we've extracted from our population recordings fail to show significant periodic structure. Beyond this, the CTC theory differs from our own in two important ways. First, oscillations target feature-selective rather than task-informative neurons[58]. These could be the same for a detection task, but differ for a discrimination such as that used in our experiment. Second, the CTC decoder uses a fixed (as opposed to a modulator-dependent) threshold, which we've shown to be suboptimal. Overall, the CTC framework describes a fixed labeling strategy based on tuning properties, while our theory proposes modulatory labeling adapted to task structure.

Some tasks, such as the context-dependent sensory evidence integration experiments by Mante and Sussillo[19], can achieve flexibility through the reorganization of late decision stages. We believe these mechanisms cannot explain flexibility in a low-level sensory discrimination task, such as the one presented here. First, numerical experiments using our hierarchical model demonstrate that it is particularly hard to achieve good performance in our task when adapting the readout alone. In addition, the recurrent dynamics supporting task switching are trained through extensive optimization[19] and although several proposals exist for the biological implementations of such learning[60], all require vast amounts of task experience. A final distinction is that our approach does not rely on an explicit context cue: the task relevance of sensory features is communicated solely through task feedback. Overall, multiple mechanisms for task-specific readout are likely to coexist in the brain and be engaged in a context dependent manner.

Our theory is agnostic to the source of the modulator and the circuit mechanisms underlying its task-specific targeting, but some previous studies provide potential clues. Changes in noise correlations across tasks could arise through either local circuit dynamics[27] or top-down mechanisms[24,61], and later propagate to downstream regions. Given the sparsity of top-down connections relative to the full population size (at least, in V1), the reorganization of noise correlations likely needs to involve local recurrent dynamics, potentially taking advantage of its topographic organization. If this kind of spatially localized modulation was indeed an organizing principle of neural activity, it would predict that flexible decoding is most effective for tasks relying on sensory features that are localized in some brain area. Consistent with this idea, Nienborg and Cumming found that V1 neurons' choice probability was significantly larger for orientation discrimination than for disparity discrimination, suggesting that V1 shows decision-related activity only if the task features are localized in the columnar organization[37]. Moreover, in a task involving higher order features, Koren et al. found neural variability was high in V4, but not V1, suggesting that the modulator could target later stages of processing depending on the task[62]. The spatial extent over which the presence of distractors may engage the V1 modulator is unclear, as the task-relevant stimuli are always placed close to one another in the experiment. However, in the additional dataset of exclusively MT recordings, stimuli were spaced further apart to accommodate the larger RFs, and the cells still exhibit modulation (Fig. 6). Furthermore, results from Rabinowitz et al. analyzing neurons in area V4 suggest that similar modulation is present in tasks with spatially distant distractors. Regarding the physiological origins of our modulator, one potential source for low-dimensional broadcast signals could be thalamic nuclei that integrate sensory and top-down information[63,64]. Alternatively, it may be possible to eliminate the need for a copy of the modulator at the readout stage, by estimating the signal directly from the observable correlations in population activity.

The lack of a biologically plausible theory of neural decoding is a fundamental shortcoming in current understanding of neural computation. Resolving the puzzle of how sensory information is routed through brain regions and extracted to perform specific tasks is critical for the study of sensory and cognitive dysfunction, including clinical applications such as brain-computer interfaces (BCI)[65]. Moreover, flexible task-dependent information routing poses a fundamental obstacle for the development of adaptive artificial intelligence systems. The framework presented here proposes a solution for this problem, supported by both physiological data and computational theory.

## Methods

### Theoretical framework for decoding from a neural population
We simulated a binary discrimination task analogous to the experiment, which requires discriminating stimuli $s = 0$ from $s = 1$ on the basis of the activity of a population of $N$ neurons. Neural responses are modeled as Poisson draws with a stimulus-dependent firing rate, which is itself modulated by a time-varying noisy signal, $m_t$, shared across neurons:

$$k_{n,t}(s,m_t) \sim \text{Poisson}\left(\lambda_n(s) \exp(c_n m_t)\right), \quad (4)$$

where $\lambda_n(s)$ is the stimulus response function of the neuron, and $t$ indexes time within a stimulus presentation. The modulator $m_t$ is 1-dimensional i.i.d. Gaussian noise with zero mean and variance $\sigma_m^2$; the nonlinearity $\exp(\cdot)$ ensures that the final firing rate is positive. The degree of modulation is neuron specific, parametrized by modulation weights $c_n$, which we take to be proportional to the $n$-th neuron's ability to discriminate the two stimuli, $c = |\log(\lambda_n(1)) - \log(\lambda_n(0))|$. We normalize responses by the expected increase in mean rate due to the modulator, $\exp\left(\frac{\sigma_m^2 c_n^2}{2}\right)$ to compensate for systematic differences in

mean firing rate due to modulation. The relative modulator strength is defined as the ratio between modulator-induced and stimulus-induced variance.

Given this modulated Poisson encoding model, an ideal observer decides the stimulus based on the sign of the log odds ratio, which reduces to comparing a weighted linear combination of the observed neural spike counts against a modulator-dependent time-varying threshold (see also ref. 12):

$$\sum_n a_n^{(\text{opt})} k_{n,t} > q^{(\text{opt})}(m_t), \quad (5)$$

with weights

$$a_n^{(\text{opt})} = \log(\lambda_n(1)) - \log(\lambda_n(0)), \quad (6)$$

and time-varying threshold

$$c^{(\text{opt})}(m_t) = -\sum_n \exp(m_t c_n)\left[\lambda_n(1) - \lambda_n(0)\right]. \quad (7)$$

The modulator-guided heuristic decoder assumes access to the modulator $m_t$ and the neural responses $k_{n,t}$, and learns approximate decoding weights based on co-fluctuations of the two within a trial:

$$\left|a_n^{(\text{MG})}\right| = \frac{1}{T} \sum_t m_t k_{n,t}. \quad (8)$$

The sign of the decoding weight is separately estimated by comparing responses to the two stimuli (trial feedback; see also ref. 12 and Supplementary Note S10).

The sign-only decoder subtracts the summed responses of two subpopulations (i.e., a linear decoder with weights ±1):

$$a_n^{(\text{SO})} = \text{sign}(\lambda_n(1) - \lambda_n(0)). \quad (9)$$

Decoders were trained on simulated data of 10,000 stimulus presentations (unless otherwise specified). To equate the amount of data available to all decoders, the modulator fluctuated at the same time scale as the stimulus.

### Hierarchical information propagation with learned stochastic modulation
We use a 4-layer artificial neural network that maps an image stimulus with 3136 pixels into categories, corresponding to 10 digits or different orientations. The first encoding layer includes neurons with fixed Gabor receptive fields.

The modulator affects encoding neurons through coupling terms $c_n$, which modulate the neuron's responses:

$$h_{n,t}^{(0)} = \exp\left(\mathbf{w}_n^{(0)} \mathbf{s} + m_t c_n\right), \quad (10)$$

where $h_{n,t}^{(0)}$ is the activity of neuron $n$ in the encoding layer, $\mathbf{w}_n^{(0)}$ are the weights from the input to this neuron. Neurons in the top layer include a multiplicative gain $g_n \geq 0$:

$$h_{n,t}^{(2)} = g_n \lfloor \left(\mathbf{w}_n^{(2)} \mathbf{h}_t^{(1)} + b_n^{(2)}\right), \quad (11)$$

where $b_n^{(2)}$ is a neuron-specific bias, optimized together with the weights $\mathbf{w}_n^{(2)}$ during pre-training. The gain $g_n$ is learned using the MG correlation rule:

$$g_n = \frac{1}{T} \sum_t m_t h_{n,t}^{(2)}(\mathbf{s}), \quad (12)$$

where $h_{n,t}^{(2)}(s)$ denotes the activity at time $t$ of neuron $n$ in the last processing layer, in response to stimulus $s$.

There are three stages of learning. (1) Pre-training optimizes all network weights to natural image statistics using a digit classification task (locally placed MNIST digits[39]) with image presentation and pixel specific i.i.d. additive Gaussian noise, L1 regularization on the weights with regularization strength 0.001 optimized for classification accuracy on a separate validation dataset of size 10,000 datapoints, while $m_t = 0$ and $g_n = 1$. (2) Learn an orientation discrimination readout from the neural responses of the fixed pretrained network (10 categories), when the input consists of single, local oriented gratings at various positions ($14 \times 14$ positions). (3) Optimize the modulator targeting for an orientation discrimination task at one fixed task location, in the presence of distractors. The task involves binary discrimination of two oriented gratings with distractor gratings at other locations. At the fast time scale $t$, the modulator varies with 100 time points per stimulus presentation, i.i.d. $m_t \sim N(0, 0.1)$, which drives gain changes in the last layer (Eq. (12)). At the slow scale (stimulus presentations) $m = 1$ and the coupling strengths $c_n$ are optimized by backpropagation.

We compare the performance of our model ("stochastic modulator", 2560 parameters for backpropagation, 7840 parameters including MG gain adjustment) to three controls: (1) full retraining of all connections ("weight retraining", 256,690 parameters, with L1 regularization as in pretraining), (2) retraining the decision layer weights ("decision layer retraining", 78,410 parameters, with L1 regularization as in pretraining), (3) all network weights are fixed, but the modulator is active $m_t = 1$, but constant, and the modulator coupling $c_n$ are optimized for the task ("deterministic modulator", 2560 parameters). In the first two approaches $m_t = 0$ and $g_n = 1$.

## Population recordings in V1 and single units from MT
In experiments by Ruff and Cohen[21], two adult male rhesus monkeys performed a motion direction change detection task on one out of 2–3 oriented drifting gratings at high or low contrast on a screen. The task-relevant grating is indicated by a few instructional stimulus presentations, selected randomly for each block within the session (3–6 blocks per session). Most recording sessions analyzed use a 10 by 10 microelectrode array (Blackrock Microsystems) in area V1 and a recording chamber with access to area MT, allowing simultaneous recordings in the two areas (multiunit activity, details in refs. [21]).

Two stimuli were positioned to drive the MT unit similarly and one stimulus was positioned outside of the MT RF. Within a block, changes in one out of the three stimuli had to be reported. In each trial, grating stimuli flash on (200 ms) and off (200–400 ms) at the same orientation (repeated, stimulus 0) until a change occurs at an unknown time (target, stimulus 1). Stimuli vary in both contrast and orientation, at each presentation, randomly interleaved. We analyzed 67 blocks of 20 recording sessions across two monkeys where the task-relevant stimulus was positioned in the RF of the population (relevant tasks) and 20 blocks of 20 sessions where the stimulus outside of the RF was task-relevant (control task). Control and relevant task blocks were interleaved within a session. Neural populations may overlap across sessions.

We analyze 21–109 trials per block, where the monkey either detected the target (hit) or failed to detect it (miss). We discard trials where the monkey did not finish the task in a hit or miss and trials where one of the distractors changed orientation. This yields an average of 54 trials per block, each with several stimulus repeats and completed by a target presentation ($s = 1$, orientation-change). We only include blocks with a minimum of 20 valid trials (77 out of 90 blocks), as numerical simulations suggest 20 trials to be the minimum necessary to estimate informativeness reliably. Varying this criterion does not qualitatively change the results. The first stimulus in a trial was always removed to eliminate adaptation effects[30]. We only include

units whose response to either one of the stimuli (presented individually) was at least 10% larger than baseline, to avoid inclusion of noise channels. On average 88 units (~90%) in a block showed stimulus modulation for one of the two stimuli placed within the MT RF (min 52, max 95). We further exclude units with a Fano factor >5 standard deviations above the population average as this suggested especially many/diverse neurons in the unit, and firing rates < 1Hz (in total 0–3 units were excluded per block).

## MT population recordings
An additional set of sessions (14 sessions with a total of 73 task blocks) in the same task had either exclusively MT recordings (24 channel probes) or simultaneous V1 and MT population recordings. The stimuli were placed to optimally drive the MT units (but not necessarily V1) with the center-to-center distance of the two adjacent stimuli typically between 2 and 3 degrees of visual angle (in contrast, in the V1 data the center-to-center distance was always around 0.5 degrees).

## Informativeness of a unit
The informativeness of a unit is quantified by $d' = \left| \frac{\mu_0 - \mu_1}{\sqrt{0.5(\sigma_0^2 + \sigma_1^2)}} \right|$ where $\mu_0$ and $\sigma_0^2$, $\mu_1$ and $\sigma_1^2$ are the means and variances of a unit's responses to the task-relevant stimulus 0 and stimulus 1, respectively. We compute informativeness across all stimulus presentations in behaviorally correct trials of the same block. Significance is assessed w.r.t. a null-distribution of $d'$ values, constructed by comparing mean and variance of random subsets of stimulus 0 responses ($p < 0.01$).

## SR model
Stimulus effects are modeled with Linear-Nonlinear Poisson (LNP), taking into account effects of repeated stimulus presentations of stimulus 0, time varying in 50ms time bins and the effects of contrast (V1) or contrast+direction (MT). Orientation is not one of the stimulus dimensions as it does not change during the repeated stimulus presentation. Responses to target stimulus 1 are used only to compute informativeness and for decoding. Stimuli are parametrized by a one-hot encoding vector with 4 time windows during 200 ms stimulus presentation; this yields 8 stimulus dimensions for the contrast-specific V1 model, with one additional dimension indicating the stimulus drift direction in MT. We add one after-stimulus dimension to capture potential delayed effects of the stimulus presentation, and an offset for base firing:

$$\boldsymbol{k}_{n,t} \sim \text{Poisson}\left(\exp\left(\beta_n \mathbf{s}_t\right)\right) \quad (13)$$

with spike counts measurements $\mathbf{k}_n$. Parameters $\boldsymbol{\beta}_n$ are obtained by maximizing the log-likelihood of the data, separately for each block:

$$L(\beta_n) = \sum_t -(\beta_n \mathbf{s}_t)^T \boldsymbol{k}_{n,t} + \exp(\mathbf{1}^T \beta_n \mathbf{s}_t) + \alpha \beta_n{}^T \beta_n. \quad (14)$$

The extended MT SR model includes the (normalized) V1 modulator as an additional predictive variable.

## Modulated SR model
We use the framework of Poisson Linear Dynamical Systems (PLDS, refs. [23,28]), to model the temporal dependencies within a trial while treating different trials as independent. The modulator terms of the PLDS are shared across the population and influence each unit's activity through a linear mapping function $\mathbf{C}$ (equivalent in meaning to

the coupling **c** in the theory). This joint model has the form:

$$\mathbf{k}_t \sim \text{Poisson}(\exp(\mathbf{Cm}_t + \mathbf{Bs}_t)) \tag{15}$$

$$\begin{aligned} \mathbf{m}_{t+1} &= \mathbf{Am}_t + \epsilon \\ \epsilon &\sim N(0, \mathbf{Q}) \\ \mathbf{m}_0 &\sim N(0, \mathbf{Q}_0) \end{aligned} \tag{16}$$

where the modulator $\mathbf{m}_t$ at time $t$ (within a trial across both stimulus presentation and inter-stimulus windows), is $D$-dimensional and the mapping $\mathbf{C}$ is $N \times D$, with latent dimensionality $D \ll N$. Parameter $\mathbf{A}$ implicitly defines the modulator's time constant ($\tau = -\frac{1}{\log(A)}$, for 1d latents), while $\mathbf{Q}$ and $\mathbf{Q}_0$ define the noise covariance of the modulator. The full model is fitted to data using the EM algorithm with a Laplace approximation for the E step (see ref. [28]); latent dimensionality is determined by model comparison ($D = 0-4$).

## Models validation and comparison

All models are 10-fold cross-validated, with model quality evaluated by (1) log-likelihood of test data (or the corresponding leave one neuron out predictions from [66] for the PLDS, averaging over latent posterior uncertainty by sampling), (2) variance explained by the model and (3) the pseudo-$R^2$ [67] which gives "the fraction of the maximum potential log-likelihood gain (relative to the null model) achieved by the tested model" $\frac{\log L(\hat{y}) - \log L(\bar{y})}{\log L(y) - \log L(\bar{y})}$, where $\hat{y}$ is the estimation of the hypothesized model and $\bar{y}$ is the null model. The null of the SR model had no stimulus-related dimensions with average firing as the only explanatory variable. The SR model served as null for the PLDS.

For a fraction of the population the SR model (30% of neurons) does not improve prediction over a constant rate model, suggesting that those neurons are not modulated by the stimulus. As expected, informative neurons show significant improvements in fit quality from the SR model relative to the null (only 5% of informative neurons do not show improvements).

## Modulator targeting

For Fig. 2G, we computed the rank of each modulator coupling in its own block-specific population and compare the distribution of significantly informative to uninformative units. In Fig. 2H–J, we used partial correlations to test for a relationship between unit's modulator coupling and task-informativeness in each block not explained by differences in overall firing rate. Specifically, we report the Spearman correlation between residual informativeness (after linearly regressing firing rate) and modulator coupling.

## Modulator strength

When assessing the overall modulation strength, both the mapping $\mathbf{C}$ and the modulator variance need to be considered jointly (as scaling up the mapping and decreasing the variance leaves results unchanged). We quantify the overall modulator strength as the variance of the modulator multiplied by the coupling norm $\sqrt{\sum_n C_n^2}$. The relative modulation strength is obtained by comparing to the stimulus drive, given by $\sum_{n,i} \text{Var}(s_i B_{n,i})$ for each neuron $n$, where $i$ indicates the stimulus dimension.

## Linking behavioral choice to neural activity

We compute the difference in target-response between trials with correct target detection and those where the monkey missed the target, normalized by their variance $\left| \frac{\mu_1 - \mu_2}{\sqrt{0.5 \cdot (\sigma_1^2 + \sigma_2^2)}} \right|$ where $\mu_{1,2}$ and $\sigma_{1,2}^2$ are the means and variances of activity corresponding to the two choices, respectively. This provides an estimate of how involved a unit was in the choice of the animal. To asses the relationship with modulator strength we use a partial correlation with two covariates, firing rate and informativeness (by multivariate linear regression).

## Decoding

We train each decoder on data that includes a balanced number of stimulus 0/1 presentations at high and low contrast, varying the size of the training set from the minimum 4 (one for each stimulus-contrast pair) to all available data (32 blocks, analyzed individually, each with 18–178 stimulus presentations, average 90). Decoder performance is tested on held out data. The optimal decoder uses maximum likelihood estimates (as in theory, with a 200ms decoding window), but based on estimated instead of ground truth parameters. It uses a constant threshold which is optimized on the training data. This is known to be suboptimal (Eq. (2)), but is more robust to the noise in the data and therefore better in the limited data regime. The modulator-guided (MG) decoder estimates readout weights by taking the inner product between the unit's activity and the modulator values (Eq. (8), using 50ms bins), with signs determined from trial-level feedback, and a constant threshold. Logistic regression learning used L2 regularization and a regularization strength of $\alpha = 1$ optimized for performance accuracy on the held-out data (10% of total available stimulus presentations in each block) [68]. SVM decoding used the off-the-shelf linear kernel SVM implementation from the *scikit-learn* library [68].

## Reporting summary

Further information on research design is available in the Nature Portfolio Reporting Summary linked to this article.

## Data availability

The data used for the analysis described in Figs. 1, 2, 4, and 6 has previously been published in ref. 20 and is available upon reasonable request. An example dataset to illustrate fitting of the PLDS model is available on figshare https://doi.org/10.6084/m9.figshare.24299131. Source data are provided with this paper.

## Code availability

The analysis was done in python and the following packages were used: numpy 1.26.0 [69], matplotlib 3.7.2 [70], scikit-learn 0.21.3 [68], pandas 2.0.3 [71], SciPy 1.3.1 [72], pickle 0.7.5 [73], Pytorch 2.0.1 [74]. The code for reproducing the modeling results in Figs. 3 and 5 and Supplementary Fig. S4 is publicly available on https://github.com/CarolineHaimerl27/modulator_guided_decoding.

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

## Acknowledgements

This project received funding from the Google PhD fellowship (C.H.), the National Science Foundation under NSF Award 1922658 (C.S), Howard Hughes Medical Institute (E.P.S., C.H.) and The Simons Foundation (E.S.). We thank Colin Bredenberg, Edoardo Balzani and David Hocker for helpful comments on the manuscript.

## Author contributions

C.H., C.S., E.P.S., D.A.R., and M.R.C. designed research. D.A.R. performed biological experiments. C.H. analyzed biological experimental data and performed simulation experiments. C.S. and E.P.S. supervised the project. C.H., C.S., and E.P.S. wrote the paper.

## Competing interests

The authors declare no competing interests.

## Ethical regulations

All animal procedures were approved by the Institutional Animal Care and Use Committees of the University of Pittsburgh and Carnegie Mellon University.
