## [Peer Review File · Nature Communications]

Targeted V1 comodulation supports task-adaptive sensory decisionsREVIEWER COMMENTS

Reviewer #1 (Remarks to the Author):

The article by Haimerl et al. studies the role of rapidly fluctuating gain modulation in facilitating the selection of the neurons that should be read out to perform a visual orientation discrimination task. They show that responses in V1 are co-modulated and that the neurons that are most strongly modulated are also the most informative. This observation is very interesting and paradoxical as the most important neurons actually exhibit the largest noise. The authors give a simple and elegant explanation: these highly variable fluctuations can help a downstream decoder select the important neurons. They then present a series of real and synthetic data analyses, based on different decoders and simulations that support their hypothesis. They finally report that the modulatory signal is also observed in MT, as predicted by the theory.

The article is well written, the idea is really interesting, and every time I wrote a note about a control the authors could do, I found it in the next section. I like the idea that fast fluctuations can help a decoder to identify the neurons that are encoding the relevant information. Furthermore, the experimental findings and their analyses are compelling.

My only criticism is about the discussion of the modulator guided (MG) decoder, that the authors suggest as a simplified model of the way downstream areas can select the relevant information. In principle, any reasonable decoder can take advantage of the fast fluctuations, and it is not clear to me that MG is really superior to other decoders. The authors show in Figure 4D that the MG decoder is slightly faster than regression. I think this is a fundamental figure, but I wonder whether the regression decoder is the right one for the comparison (I could not find details about its implementation). For example, a L1 weight regularizer might significantly speed up training. Also, SVMs might take into account the noise shape in a better way, and learn faster (especially if the SVM is trained to read out multiple time bins in every trial, in which there are independent noise samples). I think that the main idea that the fluctuations can help the decoder to converge to a good solution more rapidly is really interesting and it must be the correct interpretation of widely observed phenomena. I am just not completely convinced that the MG decoder, which relies on an explicit estimate of the correlations between the modulator and the individual neural responses, is the only way of taking advantage of the fluctuations. Additional simulations with other decoders would make the article even stronger.

Reviewer #2 (Remarks to the Author):

The study seeks to understand the neural mechanisms supporting the selection of task-relevant information in the cortical hierarchy. This is done by repurposing a 2016 dataset collected in the Cohen Lab, involving two monkeys performing an un-cued spatial attention and orientation discrimination task. The researchers hypothesize that retraining of a neural network for selecting appropriate information is rapid and cannot rely on a classical regression-based decoder. The behavioral findings from the original paper and the re-analysis suggest this retraining takes place within five trials. Recordings in V1 and MT provide the foundation for the authors' argument about the targeted modulation of task-relevant signals. A modulation term that is rapid and block-dependent is theorized and observed, particularly within the task-relevant population. The researchers suggest that this rapid modulation may serve as a task-specific "label" for the responses of task-relevant neurons, facilitating easier readouts by downstream areas. This idea contrasts with previous theories on low-dimensional covariability. The results further suggest that shared response fluctuations occur primarily for task-relevant neurons in V1, and this rapid modulation does not depend on stimulus strength or firing rate. They found that decoding is facilitated by targeted stochastic modulation, which matches the rapid behavioral flexibility. This modulation also appears to improve the speed of learning, as seen through hierarchical modeling, and suggests propagation along the decision hierarchy. The researchers observed that the fits on MT neurons were improved by a V1 modulator term and that MT estimated modulation matches V1 estimated modulation. MT stochastic modulation was also consistent with feedforward propagation. Overall, this research provides a unique theoretical perspective on the neural mechanisms involved in spatial attention and decision-making and the selection of task-relevant information, suggesting the existence of rapid, targeted modulation based on trial-by-trial covariability that serves as a "label" for task-relevant neurons. This modulation enhances decoding efficiency and correlates with behavioral flexibility. The consistency of modulation patterns between V1 and MT might also imply a feedforward propagation of this labeling system along the cortical hierarchy. In addition, the authors should be commended for a very clearly written and generally easy-to-follow manuscript despite the complex ideas and concepts presented – it was a pleasure to read. Overall, this is an important contribution to the literature that may hold relevance for other sensory modalities and decision processes but there are some concerns that would be helpful to address:

- 1) Number of informative neurons and overall generalizability: The number of neurons is only presented in Figure 3 (12 informative, 38 uninformative, and 4950 inactive), which seems to be a very low number of units, especially considering that 60% are shared between the two relevant tasks. This point may require further discussion or justification, as the authors make strong claims and form a theoretical model based on a very small set of task-informative neurons. Might it be possible to find another dataset to apply their approach? It need not recapitulate all aspects of their findings from the current dataset but could serve to increase confidence that the results described here can be generalized and do not depend on the 12 informative units from the 2016 study. If that is not possible, please expand and provide a justification for the generalizability of these findings and the limitations from only having a small number of informative units.

2) Neuronal subsets: The manuscript refers to a "modest proportion" of recorded V1 units being significantly informative (25.8% for monkey 1 and 18.4% for monkey 2). Considering the complexity of the task (spatial tuning*orientation discrimination), this doesn't seem modest, and this point may need clarification or rewording. There appears to be a contradiction regarding neuron subsets across tasks. On one hand, the manuscript states that unit informativeness is similar across the two relevant tasks, but it also suggests that different subsets of V1 neurons carry task-relevant information within each task block. Clarification and further explanation of these statements are needed. Specifically, line 98: "Across the two relevant tasks, unit informativeness is similar (61% of informative neurons are informative in both relevant tasks) because of the close proximity of the two relevant stimulus locations" and the next paragraph: "Within each task block, a different subset of V1 neurons carries task-relevant information". See line 224 and 225 for clarification and caveat.

3) Receptive Fields (RF) Closeness: The RF for the two tasks are very close, implying that the labeling modulation may only occur in the presence of closely located distractions. This raises a concern about the generalizability of the findings. It would be helpful to discuss this in the context of the modeling approaches, the results and discussion.

Minor concerns:

- Multi-unit S9: Multi-unit S9 appears to be quite important to the study and should be highlighted earlier in the manuscript for better reader understanding.
- Add citations in the first paragraph of the introduction: line 29 and line 32.
- Line 39: missing a word: "First within the traditional 'ideal observer' framework, statistically optimal decoder can be constructed from a complete description of response properties "of" the encoding population, as they pertain to the task."
- Figure 1B: since the V1 field are overlapping the schematic should reflect this
- Figure 1C: add the number of blocks in the legend or on the y-axis
- Figure 1D: add which groups are significantly different
- Line 105: refer to both S1 and Figure S1A (and center Figure S1 in the supplementary)
- The rate of stimulus presentation, trial length In line123-126 should be mentioned earlier (200ms presentation every 200-400ms)
- Line 340-341 refer to S6A after task informativeness (instead of S15 referring to S6A)
- Figure S2 subplot that should be Q is labelled G (also in the legend), for Q xlabel could be centered.
- It is unclear what is meant by overall stimulus presentations, does it refer to each individual flash or the entirety of stimulus 0 before stimulus 1? It is necessary to clarify this either early in the text or in the methods since it appears to be a crucial point on S3, the timescales are confusing (stimulus is both the flashing Gabor 200ms or the series of flashing Gabors)

Reviewer #3 (Remarks to the Author):

This is a rather exceptional paper in which a novel theoretical framework for understand a biological observation is proposed. The observation is that neurons in the visual cortex of the monkey show fast shared variability. Common slow variability has been described before, but this fast variability to my knowledge has not been previously reported. The amplitude of variability correlated with the task-relevance of the neurons as well. The authors propose a model in which the shared variability can be used to improve the sensitivity of a downstream decoder to the task-related information. Compared to a standard regression-based readout, which takes time to estimate the correct readout weights, the proposed method can be learned much more rapidly. The authors convincingly demonstrate the computational power of the proposed decoding scheme and make a good case for its biological plausibility. The mechanistic biological implementation is left somewhat open. Overall, this is a very interesting and novel conceptual paper with a potentially high impact.

*As you will see from the reports copied below, the reviewers raise important concerns. We find that these concerns limit the strength of the study, and therefore we ask you to address them with **additional work**. Without **substantial revisions**, we will be unlikely to send the paper back to review. If you feel that you are able to comprehensively address the reviewers' concerns, please provide a point-by-point response to these comments along with your revision. Please show all changes in the manuscript text file with track changes or color highlighting. If you are unable to address specific reviewer requests or find any points invalid, please explain why in the point-by-point response.*

We appreciate the positive feedback from the reviewers (“*exceptional paper*”, “*a pleasure to read*”, “*simple and elegant explanation*”, “*unique theoretical perspective*”, “*a very interesting and novel conceptual paper with a potentially high impact*”). The most significant criticisms seem to be about the exact **form of the readout** (R1) and the **statistical power** of the reported experimental effects (R2). To address the first, we've included additional numerical simulations and data analysis comparing the readout with alternative decoders, making explicit the benefits of the modulator-dependent solution. The second concern is due to a simple **misunderstanding**: the reviewer interpreted the numbers corresponding to a model simulation in Fig.3 (12 out of 5000 task informative neurons) as a description of the experimental dataset statistics. The model simulations were intentionally performed under this extreme scenario to emphasize the power of modulatory labeling in a regime of sparse task-informativeness. Moreover, the use of a small number of task-relevant neurons does not imply weak statistical power since all data analyses are based on the entire set of neurons (not just the task-relevant subpopulation). We have now included a table that specifies the number of neurons of various types and the amount of data supporting each of the reported results, for both the previously published dataset (Ruff et al. 2016) and the unpublished dataset (MT population recordings). Given these numbers (many thousands of trials, **>1500 informative V1 units**) and the fact that the effects are consistent across both datasets, we don't think inclusion of data from other tasks is warranted or necessary.

Detailed point-by-point responses are provided below. We believe that we have addressed all reviewer concerns and look forward to further feedback.

Reviewer #1 (Remarks to the Author):

*The article by Haimerl et al. studies the role of rapidly fluctuating gain modulation in facilitating the selection of the neurons that should be read out to perform a visual orientation discrimination task. They show that responses in V1 are co-modulated and that the neurons that are most strongly modulated are also the most informative. This observation is very interesting and paradoxical as the most important neurons actually exhibit the largest noise. The authors give a **simple and elegant explanation**: these highly variable fluctuations can help a downstream decoder select the important neurons. They then present a series of real and synthetic data analyses, based on different decoders and simulations that support their hypothesis. They finally report that the modulatory signal is also observed in MT, as predicted by the theory.*

My only criticism is about the discussion of the modulator guided (MG) decoder, that the authors suggest as a simplified model of the way downstream areas can select the relevant information. In principle, any reasonable decoder can take advantage of the fast fluctuations, and it is not clear to me that MG is really superior to other decoders.

The nature of the decoder is indeed important. While most sensible decoders would be sensitive to the covariance structure of the neural responses, the uniqueness of our proposed decoder is that it treats the fluctuating modulation as part of the **signal** rather than just noise. Aside from this key distinction, the construction of our solution is somewhat heuristic (not guaranteed to be optimal), but sensible in that 1) it has the same general form as the mathematically-derived optimal solution, 2) it is built around a weighted linear combination of afferents, a form generally assumed to be biologically plausible and 3) its weights can be estimated locally, which makes efficient learning implementable by synaptic plasticity or similar mechanisms. We do not claim uniqueness, but we do claim that decoders typically used in neuroscience do not have these properties.

As suggested, we now include a comparison of our MG decoder to **two additional decoders**: an L1 regularized linear decoder, and linear kernel SVM (see new section **S10** “Comparison to other decoders”, and new panels in Suppl. Fig. **S4H-K**) across a range of modulation amplitudes and dataset sizes. We found that:

- Performance of both decoders (for a fixed training dataset size of 10k datapoints) monotonically decreases with increasing modulation strength, similar to the optimal decoder, but unlike the inverse U shape found with our MG decoder. This reflects the systematically negative effects of modulation on neural encoding. Thus, although these new decoders are both sensitive to second order moments in the data, they are not able to exploit the information conveyed by that structure in the same way, because the modulator-induced fluctuations are treated as noise (see new Suppl. Fig. 4H,I, dark blue and purple vs. green).
- A subtle issue arises in setting the strength of regularization in the biological context. Since the operations are done online, we reasoned that regularization strength should be fixed rather than changing as more data becomes available. With the strength of L1 regularization chosen to optimize cross-validated accuracy, we find that the test performance of regression is close to the asymptotically optimal performance (see Suppl. Fig. S4H dark blue regression vs. red optimal). However, performance with limited data is very poor, presumably due to underfitting (Fig.4I,J,K). Reducing the strength of regularization can improve performance in the low data regime, but at the cost of a decrease in generalization performance for large data (Fig.4J,K), due to large variance/overfitting. In a machine learning setting, one might choose to adaptively re-optimize the regularization strength depending on the amount of data, but this seems implausible as a biological solution.
- As R1 intuited, SVM does perform well in the low data regime, but this comes at the cost of not reaching optimal performance in a larger dataset (See new Suppl. Fig. 4 H&I, purple). SVMs tend to have low bias and high variance, so behave similarly to

weak L1 regularization. Furthermore, it is unclear how the brain could implement something like SVM decoding.

Overall, these results show that conventional decoders can work well in the statistical regime for which they have been optimized for but generalize less well to other data regimes. In contrast, our heuristic is close to optimal with enough data, fast to learn from limited data, and requires no hyperparameter tuning.

For a direct dissection of the role of treating the modulation as signal instead of noise, we also considered the scenario where the **modulator is known** and fed as an additional input to the L1 regularized linear and SVM decoders (trained on 10000 data points with relative modulator strength ~ 1 , 20 independent runs of learning). This in principle would allow a modulator dependent decision threshold (as in the optimal solution). Somewhat unexpectedly, the performance of the decoders that use the modulator as input is statistically indistinguishable from the original decoders without it (logistic regression test accuracy was 86% +/- sd=0.93 when including the modulator as input vs. 86% +/- 0.93 without, paired t-test p-val = 0.85; for SVM 77% +/- 1.29 with vs. 77% +/- 1.32 without, p-val=0.89), which suggests that it's not just a matter of knowing the modulator, but using it in an effective way for learning.

Finally, the benefits of the modulation-based decoder are preserved when taking into account the propagation of information across brain areas: L1 regression (which corresponds to retraining of weights in results section 2.5 and Fig.5) performs worse than modulator-based gain tuning in the hierarchical setting. We clarified the procedure used for regression in the hierarchical setting in the methods section, lines 502, 513-514.

Overall the MG decoder is qualitatively different from traditional solutions in 1) its speed-accuracy flexibility, 2) the notion of an optimal noise level for effective decoding, and 3) biological plausibility of the readout and its weight estimation.

Summary of manuscript changes:

- New supplementary subsection "Comparison to other decoders" in section S10 including two additional decoders.
- Results summarized in Suppl. Fig. 4, new panels H-K
- New results mentioned in main text line 212-214

The authors show in Figure 4D that the MG decoder is slightly faster than regression. I think this is a fundamental figure, but I wonder whether the regression decoder is the right one for the comparison (I could not find details about its implementation).

Thanks for noting - we have clarified the missing details in the methods section "Decoding", lines 587.

*For example, a **L1 weight regularizer** might significantly **speed up training**. Also, **SVMs** might take into account the noise shape in a better way, and learn faster (especially if the SVM is trained to read out multiple time bins in every trial, in which there are independent noise samples).*

Regarding the potential discrepancy in the time scale at which data reaches the modulation (i.e. slow for trial time scale vs. fast for the modulator time scale) our new numerical simulations match the time scale of stimulus and modulator changes so all decoders operate on the same time scale (**Suppl. Fig.S4H-K**). The MG benefits for learning remain intact. There is however a conceptual distinction between the supervision signal that arrives at the slow (trial-scale) rate and the fast modulation; this corresponds to a factor of 4 in our data analysis, but is potentially larger from the perspective of the brain. Furthermore, in the real data the fast time scale samples are strictly speaking not independent, even if the autocorrelation time scale of the modulator is quite fast.

Given the results on the simulated data, we decided to also include the SVM decoder in our data decoding analysis (see **new Suppl. Section S15** “SVM decoding from data”). Unlike the theory, the SVM decoder has sample efficiency similar to our MG decoder. This apparent discrepancy relative to the previous simulations has to do with the fact that the modulator is no longer directly observed, but rather estimated via a maximum likelihood fit of the Poisson LDS model. Nonetheless, the point of this analysis is mainly to show that the heuristic is still competitive, despite the distinctions between what information is experimentally available vs what is accessible to the brain (see also discussion about biases in experimental sampling and how they lead to underestimating the role of modulation). Associated main text changes in lines 266-269.

*I think that the main idea that the fluctuations can help the decoder to converge to a good solution more rapidly is really interesting and it must be the correct interpretation of widely observed phenomena. I am **just not completely convinced that the MG decoder, which relies on an explicit estimate of the correlations between the modulator and the individual neural responses, is the only way of taking advantage of the fluctuations.** Additional simulations with other decoders would make the article even stronger.*

We appreciate the comments/suggestions, and believe that the addition of the new decoders has helped sharpen the message of the article.

Reviewer #2 (Remarks to the Author):

The study seeks to understand the neural mechanisms supporting the selection of task-relevant information in the cortical hierarchy. This is done by repurposing a 2016 dataset collected in the Cohen Lab, involving two monkeys performing an un-cued spatial attention and orientation

discrimination task. The researchers hypothesize that retraining of a neural network for selecting appropriate information is rapid and cannot rely on a classical regression-based decoder.

*The behavioral findings from the original paper and the re-analysis suggest this retraining takes place within five trials. Recordings in V1 and MT provide the foundation for the authors' argument about the targeted modulation of task-relevant signals. A modulation term that is rapid and block-dependent is theorized and observed, particularly within the task-relevant population. The researchers suggest that this rapid modulation may serve as a task-specific "label" for the responses of task-relevant neurons, facilitating easier readouts by downstream areas. This idea contrasts with previous theories on low-dimensional covariability. The results further suggest that shared response fluctuations occur primarily for task-relevant neurons in V1, and this rapid modulation does not depend on stimulus strength or firing rate. They found that decoding is facilitated by targeted stochastic modulation, which matches the rapid behavioral flexibility. This modulation also appears to improve the speed of learning, as seen through hierarchical modeling, and suggests propagation along the decision hierarchy. The researchers observed that the fits on MT neurons were improved by a V1 modulator term and that MT estimated modulation matches V1 estimated modulation. MT stochastic modulation was also consistent with feedforward propagation. Overall, **this research provides a unique theoretical perspective on the neural mechanisms involved in spatial attention and decision-making and the selection of task-relevant information**, suggesting the existence of rapid, targeted modulation based on trial-by-trial covariability that serves as a "label" for task-relevant neurons. This modulation enhances decoding efficiency and correlates with behavioral flexibility. The consistency of modulation patterns between V1 and MT might also imply a feedforward propagation of this labeling system along the cortical hierarchy. In addition, **the authors should be commended for a very clearly written and generally easy-to-follow manuscript despite the complex ideas and concepts presented – it was a pleasure to read**. Overall, this is an **important contribution to the literature** that may hold relevance for other sensory modalities and decision processes but there are some concerns that would be helpful to address:*

*1) **Number of informative neurons and overall generalizability:** The number of neurons is only presented in Figure 3 (12 informative, 38 uninformative, and 4950 inactive), which seems be a very low number of units, especially considering that 60% are shared between the two relevant tasks. This point may require further discussion or justification, as the authors make strong claims and form a theoretical model based on a **very small set of task-informative neurons**. Might it be possible to find **another dataset** to apply their approach? It need not recapitulate all aspects of their findings from the current dataset but could serve to increase confidence that the results described here can be generalized and do not depend on the 12 informative units from the 2016 study. If that is not possible, please expand and provide a justification for the generalizability of these findings and the limitations from only having a small number of informative units.*

There is a fundamental misunderstanding about numbers here - perhaps because we did not provide a more precise description of the neural datasets. Figure 3 refers to simulations, not data. We've intentionally chosen a small number of informative neurons to illustrate the

biological challenge of extracting task relevant information from a “sea” of irrelevant activity. We believe this reflects biological reality, as most of V1 neural responses are irrelevant in our task, but not the experimental dataset, which is targeted towards a task relevant subset of neurons (see also Discussion about experimental biases and how they understate the importance of task informative subset selection).

We added a supplementary table that makes all the unit and trials numbers explicit for each individual analysis (see **Suppl. Table S1**, linked from main text line 102 and reproduced below). The main message conveyed by these numbers is that the analysis is **not data limited**. The two datasets (combining both published and unpublished data) include thousands of trials and more than **1500 informative units**. Moreover, most analyses include the full set of units, not just the informative subset.

It would be great to be able to validate the theory in the context of other tasks, however we’re not aware of any suitable data. Testing our hypothesis requires population recordings (to be able to estimate the modulator) including both task relevant and task irrelevant neurons and explicit task switching in which the animal dynamically and quickly adapts to new task contexts. Testing the propagation of information across areas also requires multiarea recordings. To our knowledge, **no publicly available dataset satisfies these requirements**. We do hope that this paper will encourage future experiments that can address the role of modulation for behavioral adaptation more fully.

	Fig. 1	Fig. 2	Fig. 3	Fig. 4	Fig. 5	Fig. 6
V1 trials/block	3640	3640/2348 (G-J)		2348		
V1 neural units/block	5903	5903/4004 (G-J)		4004		
Informative units V1	1553					
Single MT units paired with V1 population recordings						13* (A, C, D)
MT units in population recordings						1752 (E) 96 (F)
Simulated Poisson population			12/38/4950 inform. /uninforma tive			

			/inactive			
Simulated image responses					2560 in encoding layer	

* excludes sessions for which the modulated SR model was not a good fit for the V1 population

List of changes:

- New Table in Suppl. that lists the precise amount of trials & neural units that was used for each analysis in the main text figures.
- Reference in main text lines 102

*2) Neuronal subsets: The manuscript refers to a "modest proportion" of recorded V1 units being significantly informative (25.8% for monkey 1 and 18.4% for monkey 2). Considering the complexity of the task (spatial tuning*orientation discrimination), this doesn't seem modest, and this point may need clarification or rewording.*

We apologize for the lack of clarity. While this may seem like a significant percentage of informative cells, it is not if one takes into account the experimental bias. Since the task relevant stimulus is specifically optimized to drive this population, an 18-25% fraction of informative cells seems modest. We've edited the main text to explain this in **line 97**.

There appears to be a contradiction regarding neuron subsets across tasks. On one hand, the manuscript states that unit informativeness is similar across the two relevant tasks, but it also suggests that different subsets of V1 neurons carry task-relevant information within each task block. Clarification and further explanation of these statements are needed. Specifically, line 98: "Across the two relevant tasks, unit informativeness is similar (61% of informative neurons are informative in both relevant tasks) because of the close proximity of the two relevant stimulus locations" and the next paragraph: "Within each task block, a different subset of V1 neurons carries task-relevant information". See line 224 and 225 for clarification and caveat.

We apologize for the lack of clarity. We have updated the text accordingly (lines 104).

3) Receptive Fields (RF) Closeness: The RF for the two tasks are very close, implying that the labeling modulation may only occur in the presence of closely located distractors. This raises a concern about the generalizability of the findings. It would be helpful to discuss this in the context of the modeling approaches, the results and discussion.

It is indeed the case that our task-relevant condition always includes a nearby distractor, which makes it difficult to make strong statements about the conditions under which the modulator-based mechanism should be engaged. However, part of the unpublished data includes stimuli that were spaced further apart (2-3 degrees of visual angle instead of 0.5 degrees) and modulation was still extracted from the neural population. Finally, similar task-dependent modulation has been previously reported in Rabinowitz et al. (2015), under

conditions where the distractor is quite far from the target (in the other hemifield). Taking these observations together, we expect to see modulation across a range of distractor distances.

We made the following changes following the reviewer comment:

Changes in the discussion, see line 463-468:

“The spatial extent over which the presence of distractors may engage the V1 modulator is unclear, as the relevant task stimuli are always placed close to one another in the V1 experiment. However, in the additional dataset of exclusively MT recordings, the stimuli were spaced further apart to accommodate the larger MT RFs. We still find modulation in the MT population (see Fig. 6). Further, results from Rabinowitz et al. (2015) analyzing neurons in area V4 suggest that similar modulation is present in tasks with spatially distant distractors.”

Changes in the methods, see line 547-549:

“The stimuli were placed to optimally drive the MT units (but not necessarily V1) with the center-to-center distance of the two adjacent stimuli typically between 2 and 3 degrees of visual angle (in contrast, in the V1 data the center-to-center distance was always around 0.5 degrees).”

Minor concerns:

- Multi-unit S9: Multi-unit S9 appears to be quite important to the study and should be highlighted earlier in the manuscript for better reader understanding.

Added sentence in line 99: “Impact of multiunits is discussed in Suppl.S9.”

- Add citations in the first paragraph of the introduction: line 29 and line 32.

Added references to Born et al, 2005, Rust et al, 2010 and Hubel&Wiesel, 1959.

- Line 39: missing a word: “First within the traditional ‘ideal observer’ framework, statistically optimal decoder can be constructed from a complete description of response properties “of” the encoding population, as they pertain to the task.”

Corrected.

- Figure 1B: since the V1 field are overlapping the schematic should reflect this

Given the large number of V1 units, we were concerned with overcrowding the figure by explicitly adding individually mapped V1 RFs. The size of the RFs is indicated by the circle marking the stimulus locations, since according to the experimental design, the stimulus size is matched to the typical RF size at that eccentricity in V1. We clarified this in the figure caption:

“Two of the three stimuli locations are within the MT unit’s receptive field (“relevant” - purple circle, matched to average V1 RF size) and one is in the opposite hemifield (“control” - black).”

- Figure 1C: add the number of blocks in the legend or on the y-axis
Number added in figure caption.

- Figure 1D: add **which groups** are significantly different

The caption already indicated that “the red star indicates a significant difference in means”
Added: “ $p=0.015$ for comparison between <5 and <10 groups, $p>0.05$ for other comparisons”

- Line 105: refer to both S1 and Figure S1A (and center Figure S1 in the supplementary)
Done.

- The rate of stimulus presentation, trial length In line 123-126 should be mentioned earlier
(200ms presentation every 200-400ms)

Edited **Line 66**: “Two to three gratings were present simultaneously, at high or low contrast levels, and spontaneously changed their orientation after a varying number of repeats (stimulus on for 200ms, off for 200-400ms)”

- Line 340-341 refer to S6A after task informativeness (instead of S15 referring to S6A)
Corrected.

- Figure S2 subplot that should be Q is labeled G (also in the legend), for Q xlabel could be centered.

Corrected.

- It is unclear what is meant by overall stimulus presentations, does it refer to each individual flash or the entirety of stimulus 0 before stimulus 1? It is necessary to clarify this either early in the text or in the methods since it appears to be a crucial point on S3, the timescales are confusing (stimulus is both the flashing Gabor 200ms or the series of flashing Gabors)

Stimulus always refers to a single flashing of a Gabor stimulus. We now clarified this in the main text and the experimental description in the methods (see changes in “Population recordings in V1 and single units from MT” lines 525, 527)

Reviewer #3 (Remarks to the Author):

This is a rather exceptional paper in which a novel theoretical framework for understanding a biological observation is proposed. The observation is that neurons in the visual cortex of the monkey show fast shared variability. Common slow variability has been described before, but this fast variability to my knowledge has not been previously reported. The amplitude of variability correlated with the task-relevance of the neurons as well. The authors propose a model in which the shared variability can be used to improve the sensitivity of a downstream decoder to the task-related information. Compared to a standard regression-based readout, which takes time to estimate the correct readout weights, the proposed method can be learned much more rapidly. The authors convincingly demonstrate the computational power of the

*proposed decoding scheme and make a good case for its biological plausibility. The mechanistic biological implementation is left somewhat open. Overall, this is **a very interesting and novel conceptual paper with a potentially high impact.***

We appreciate the positive feedback. Regarding mechanistic implementation, we do agree that there's challenging work ahead. We have made an effort to investigate the ability of local plasticity mechanisms to implement the readout in simulations (Suppl. Section S11 and Suppl. Fig. 4LM), which speaks to the biological plausibility of the readout. But the mechanisms underlying the generation and targeting of the modulatory signal are unknown, and not constrained by any data that we're aware of, as we mention in the discussion. We hope our paper encourages future development of experiments to target these questions more directly.

REVIEWERS' COMMENTS

Reviewer #1 (Remarks to the Author):

The authors took seriously all my comments and suggestions, and addressed all my concerns. It's a very interesting article and I believe that it is ready for publication.

Reviewer #2 (Remarks to the Author):

I'm completely satisfied with the revisions - thank you for the clarification regarding Figure 3. Given my misunderstanding, I agree that there is no need to include data from another task.

Congratulations on an excellent manuscript!